# Identification and validation of serum metabolite biomarkers for endometrial cancer diagnosis

Wanshan Liu [1,2,3,4,5], Jinglan Ma [1,2,5], Juxiang Zhang[1,2,3,4], Jing Cao[1,2,3,4], Xiaoxiao Hu[1,2], Yida Huang [1,2,3,4], Ruimin Wang[1,2,3,4], Jiao Wu[1,2,3,4], Wen Di [1,2✉], Kun Qian [1,2,3,4✉] & Xia Yin [1,2✉]

## Abstract

**Endometrial cancer (EC) stands as the most prevalent gynecological tumor in women worldwide. Notably, differentiation diagnosis of abnormity detected by ultrasound findings (e.g., thickened endometrium or mass in the uterine cavity) is essential and remains challenging in clinical practice. Herein, we identified a metabolic biomarker panel for differentiation diagnosis of EC using machine learning of high-performance serum metabolic fingerprints (SMFs) and validated the biological function. We first recorded the high-performance SMFs of 191 EC and 204 Non-EC subjects via particle-enhanced laser desorption/ionization mass spectrometry (PELDI-MS). Then, we achieved an area-under-the-curve (AUC) of 0.957–0.968 for EC diagnosis through machine learning of high-performance SMFs, outperforming the clinical biomarker of cancer antigen 125 (CA-125, AUC of 0.610–0.684, $p < 0.05$). Finally, we identified a metabolic biomarker panel of glutamine, glucose, and cholesterol linoleate with an AUC of 0.901–0.902 and validated the biological function in vitro. Therefore, our work would facilitate the development of novel diagnostic biomarkers for EC in clinics.**

**Keywords** Biomarker; Endometrial Cancer; Machine Learning; Mass Spectrometry; Metabolite
**Subject Categories** Biomarkers; Cancer; Urogenital System

## Introduction

Endometrial cancer (EC) stands as the most prevalent gynecological tumor in women worldwide (Dou et al, 2020), with an estimated 420,000 new cases and 97,000 deaths annually (Sung et al, 2021). The endometrioid is the main histotype in EC, accounting for ~80% of patients (Lu and Broaddus, 2020). Notably, the 5-year survival rate of early EC was > 90%, as opposed to a 15–74% survival rate for advanced or metastatic EC (Crosbie et al,

2022; Lu and Broaddus, 2020). Therefore, timely differentiation diagnosis of EC and Non-EC in abnormity detected by ultrasound findings (e.g., thickened endometrium or mass in the uterine cavity) is essential for optimal patient outcomes (Jones et al, 2021; Koskas et al, 2021). However, existing diagnostic tools, including transvaginal ultrasound, biopsy, and curettage, are limited by low specificity (~51.1% for transvaginal ultrasound at the endometrial thickness cut-off of 5 mm) or invasiveness (endometrial sampling for biopsy and curettage) (Jones et al, 2021). Further, the commonly used blood biomarker for EC diagnosis in clinics is cancer antigen 125 (CA-125), limited by its low sensitivity (<60%) (Njoku et al, 2019). Thus, there is an urgent need for alternative biomarkers in the blood to enable timely differentiation diagnosis of EC and Non-EC for potential clinical use.

Emerging blood-based biomarker assays are promising for large-scale cancer diagnostics, focusing on genes, proteins, and metabolites (Cheng et al, 2019; El-Deiry et al, 2019; Yao et al, 2023b). Particularly, metabolite biomarkers offer a more direct snapshot of disease phenotype than gene and protein biomarkers (Buergel et al, 2022; Yang et al, 2022). Moreover, metabolic reprogramming is recognized as a hallmark of malignancy, and the reprogrammed metabolic activities can be exploited for diagnostic purposes (Faubert et al, 2020). However, the current metabolite biomarkers in blood for EC diagnosis remain limited, given the traditional analytical techniques and lack of validated biological function (Ihata et al, 2014; Knific et al, 2018; Troisi et al, 2018; Yan et al, 2022). Therefore, there is a need to construct a metabolic biomarker panel based on advanced analytical techniques with a designed EC cohort and validate its biological function in vitro.

Mass spectrometry (MS) represents the principal technique in metabolite detection, offering high sensitivity, resolution, and identification capability in a label-free manner by measuring the accurate mass-to-charge ratio ($m/z$) (Aron et al, 2022; Bakker et al, 2022). However, metabolite detection in complex bio-fluids for MS requires deproteinization and liquid/gas chromatography (LC/GC) to purify and enrich metabolites, limiting the analytical speed and capacity (Dunn et al, 2011). Recently, particle-enhanced laser desorption/ionization MS (PELDI-MS) has demonstrated enhanced analytical speed and capacity owing to the defined particles for recognition and trapping of metabolites (Huang et al, 2022; Wang

[1]Department of Obstetrics and Gynecology, Renji Hospital, School of Medicine, Shanghai Jiao Tong University, Shanghai 200127, P. R. China. [2]State Key Laboratory of Systems Medicine for Cancer, Shanghai Key Laboratory of Gynecologic Oncology, Shanghai 200127, P. R. China. [3]School of Biomedical Engineering, Institute of Medical Robotics and Shanghai Academy of Experimental Medicine, Shanghai Jiao Tong University, Shanghai 200030, P. R. China. [4]Division of Cardiology, Renji Hospital, School of Medicine, Shanghai Jiao Tong University, Shanghai 200127, P. R. China. [5]These authors contributed equally: Wanshan Liu, Jinglan Ma. ✉E-mail: diwen@renji.com; k.qian@sjtu.edu.cn; yinxia@renji.com

et al, 2022a; Wang et al, 2022b). To date, serum metabolic fingerprints (SMFs) by PELDI-MS have covered several major diseases for advanced biomedical applications and are promising in EC diagnosis (Huang et al, 2022; Pei et al, 2023; Wang et al, 2022a). Further, given the wealth of information provided by SMFs, machine learning has played an instrumental role in mapping the SMFs to clinical outcomes.

Herein, we identified a metabolic biomarker panel for differentiation diagnosis of EC using machine learning of high-performance SMFs and validated the biological function (Fig. 1). Firstly, we recorded the high-performance SMFs of 191 EC and 204 Non-EC subjects via PELDI-MS (Fig. 1A). Then, we achieved an area-under-the-curve (AUC) of 0.957–0.968 for EC diagnosis through machine learning of high-performance SMFs, outperforming the clinical biomarker of CA-125 (AUC of 0.610–0.684, $p < 0.05$, Fig. 1B). Further, we identified a metabolic biomarker panel of glutamine, glucose, and cholesterol linoleate (Fig. 1C) with an AUC of 0.901–0.902 for EC diagnosis. Finally, we validated the effect of the three metabolite biomarkers on EC cell behaviors, including proliferation, colony formation, migration, and apoptosis in vitro (Fig. 1D). Therefore, our work would facilitate the development of novel diagnostic biomarkers for EC in clinics.

# Results

## High-performance detection of metabolites via PELDI-MS

We developed an on-chip microarray using ferric oxide particles for high-performance detection of metabolites via PELDI-MS (Fig. 2A–C; Appendix Fig. S1A–F). Our approach offered several advantages, including: (1) high salt and protein tolerance with enhanced intensities (from $2.0 \times 10^2$–$1.0 \times 10^4$ to $4.1$–$4.4 \times 10^5$), (2) high reproducibility (coefficients of variation (CVs) of 5.6–11.0%) with a good linear response ($R^2 = 0.963$–$0.986$), and (3) fast analytical speed of ~30 s per sample with a high capacity of 384 samples per chip.

To investigate the high salt and protein tolerance with enhanced intensities of PELDI-MS, we included a standard sample containing five typical metabolites (alanine, proline, glutamic acid, glucose, and lactose). We observed an enhanced sum of peak intensities ($4.1$–$4.4 \times 10^5$, $p < 0.05$) for metabolites under both high salt (20 mM of $Na^+$) and biofluid-mimic (20 mM of $Na^+$ and 10 mg/mL of protein) conditions in PELDI-MS (Fig. 2D,E; Appendix Fig. S2A,B, and Appendix Table S1). In contrast, the peak intensities were significantly suppressed ($2.0 \times 10^2$–$1.0 \times 10^4$, $p < 0.05$) in LDI-MS with organic matrices (Fig. 2D,E and Appendix Table S1), due to unfavored self-ionization of organic matrices (Appendix Fig. S2C–F). We also observed enhanced intensities in PELDI-MS when analyzing serum samples (Fig. 2F), resulting in a higher total ion count (~$2.2 \times 10^7$, $p < 0.05$), as compared to organic matrices (total ion count of ~$3.2 \times 10^5$–$4.4 \times 10^6$). Therefore, the high salt and protein tolerance of PELDI-MS enabled direct detection of metabolites in complex bio-samples with enhanced intensities.

For high reproducibility with a good linear response of PELDI-MS, the CVs of peak intensities for the 5 typical metabolites were 5.6–11.0% (Appendix Table S2). Similarly, the median CVs for peak intensities of $m/z$ features in serum samples were 9.4–12.3%, and 95.1–99.2% of $m/z$ features showed CVs within 30% (Fig. 2G). The low CVs can be attributed to the homogeneous co-crystallization of particles (arithmetic mean height = 0.24 μm, Fig. 2H; Appendix Fig. S3A), in contrast to organic matrices (arithmetic mean height = 0.90–3.18 μm, $p < 0.05$, Fig. 2H and Appendix Fig. S3B–E). Notably, the high reproducibility of PELDI-MS offered a good linear response ($R^2 = 0.963$–$0.986$) with a limit of detection (LOD) of 0.41–0.53 μM in metabolite analysis (Fig. 2I and Appendix Fig. S3F,G). Furthermore, the PELDI-MS afforded a fast analytical speed of ~30 s per sample with a high capacity of up to 384 samples per chip (arranged in a configuration of 16 rows by 24 columns), allowing the automatic detection of metabolites for potential clinical applications.

## Cohort design and SMFs characterization of EC and Non-EC

In the case-control study design, serum samples of 1726 subjects were collected in a gynecological disease biobank at Renji Hospital, School of Medicine, Shanghai Jiao Tong University from Dec. 2018 (Fig. 3A). Of these, 395 serum samples of 191 EC subjects and 204 Non-EC subjects were included for further analysis. Notably, the EC and Non-EC subjects were diagnosed by two experienced pathologists independently. Clinical characteristics, like age at pathological diagnosis, level of CA-125, BMI, diabetes, menopause, hypertension, International Federation of Gynecology and Obstetrics (FIGO) 2018 stage, and histology type, are summarized (Appendix Table S3). Notably, 88.5% of subjects in EC were endometrioid histotypes, indicating our further analysis was primarily relevant to this specific histotype. This study adhered strictly to the principles of the Declaration of Helsinki and the Department of Health and Human Services Belmont Report, with the approval by the ethics committee of Renji Hospital, School of Medicine, Shanghai Jiao Tong University (2018-114).

Further, we constructed an SMFs database of the 191 EC and 204 Non-EC subjects via PELDI-MS. In the typical raw mass spectra obtained from EC and Non-EC serums, we observed strong $m/z$ features (Fig. 3B), owing to the sufficient ionization efficiency of ferric oxide particles. Accordingly, we obtained 272 $m/z$ features (defined as SMFs) using data processing (e.g., peak detection and alignment, Appendix Fig. S4A) and built the blueprint of SMFs for the 191 EC and 204 Non-EC subjects (Fig. 3C). Notably, the frequency of similarity score > 0.85 reached 96.3% and 94.6% for EC and Non-EC groups, demonstrating the similarity of SMFs within the same group (Fig. 3D). Further, by unsupervised analysis (e.g., principal component analysis (PCA)), the SMFs between EC and Non-EC groups displayed a certain degree of overlap (Fig. 3E; Appendix Fig. S4B,C), indicating the need for advanced machine learning algorithms to map the SMFs to clinical outcomes.

## Machine learning of SMFs for EC diagnosis

We performed machine learning for EC diagnosis, using optimized algorithms to study the diagnostic performance of SMFs and compared it with the CA-125. Initially, we conducted a power analysis on a preliminary study (6/6, EC/Non-EC) to determine the sample number required for statistically significant machine learning (Appendix Fig. S5A). The power analysis showed that a sample number of 200 (100/100, EC/Non-EC) can achieve a predicted power of 0.85, indicating the statistical confidence level of

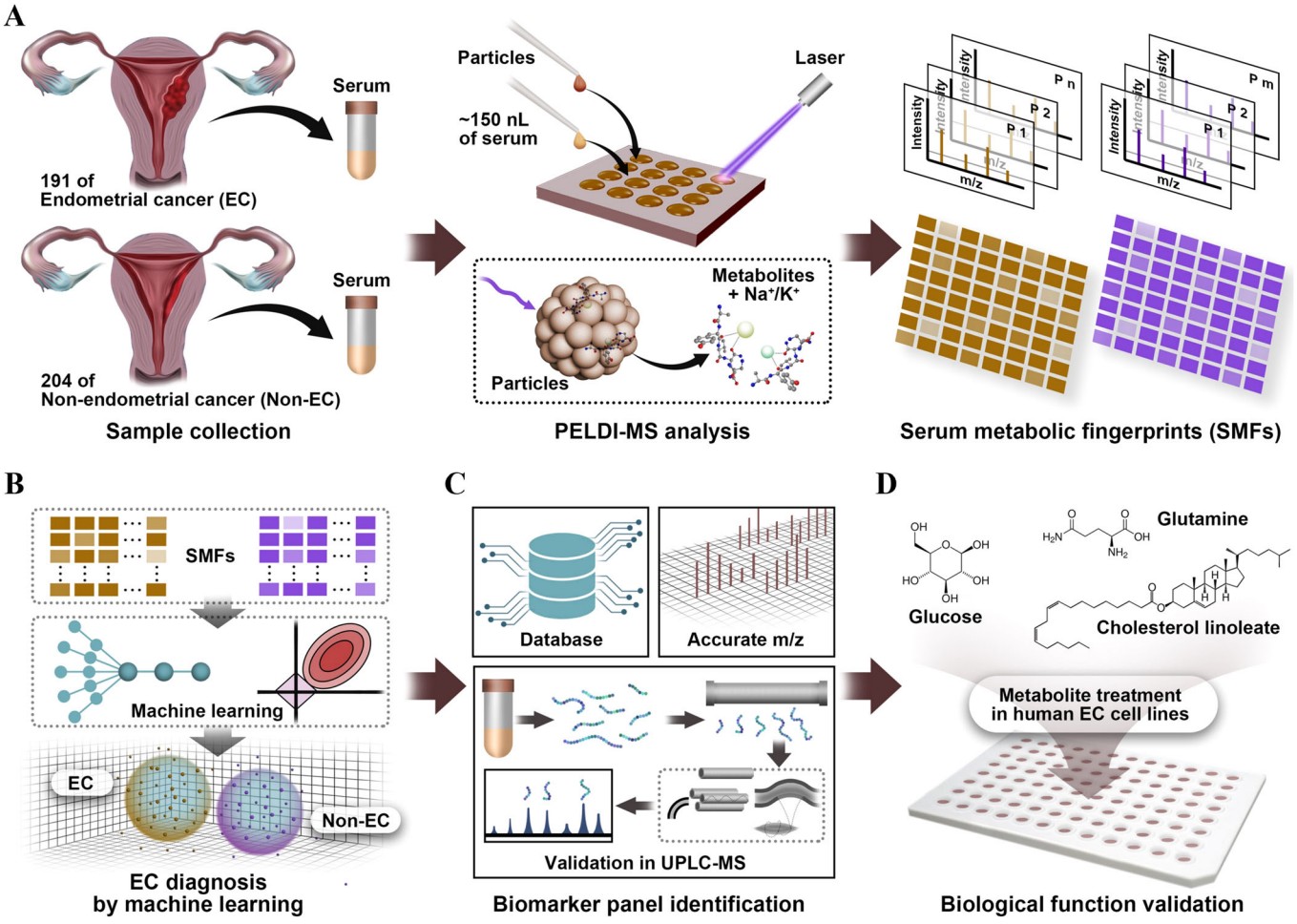

**Figure 1. Schematics for biomarker panel identification and validation.**

(A) We collected serum samples of 395 subjects (191 endometrial cancer (EC) and 204 Non-endometrial cancer (Non-EC)) and recorded the high-performance serum metabolic fingerprints (SMFs) via particle-enhanced laser desorption/ionization mass spectrometry (PELDI-MS) analysis. (B) Then, we achieved EC diagnosis by machine learning of high-performance SMFs. (C) Further, we identified a metabolic biomarker panel of glutamine, glucose, and cholesterol linoleate through accurate *m/z* and database search and validated in ultra-performance liquid chromatography-MS (UPLC-MS). (D) Finally, we validated the effect of the three metabolite biomarkers on EC cell behaviors in vitro.

the machine learning results. Next, 238 subjects (115/123, EC/Non-EC) were included in the discovery cohort to perform model optimization using fivefold cross-validation (Fig. 4A). Notably, we have matched the age distribution ($p > 0.05$) between the EC and Non-EC groups in the discovery cohort to mitigate age bias during the training phase and panel development. Finally, we included an independent validation cohort (76/81, EC/Non-EC) to evaluate the reliability and no overfitting of the optimal model (Fig. 4A).

We included five commonly used machine learning algorithms (least absolute shrinkage and selection operator (LASSO), logistic regression, partial least squares discriminant analysis (PLS-DA), random forest, and decision tree). All algorithms achieved an AUC ≥ 0.75 in the discovery cohort, demonstrating the potential of SMFs in EC diagnosis. Specifically, LASSO afforded the best performance ($p < 0.05$) with an AUC of 0.957 (95% confidence interval (CI) of 0.906–1.000, Fig. 4B), compared to the other machine learning algorithms (AUC of 0.940 and 95% CI of 0.876–0.998 for logistic regression, AUC of 0.940 and 95% CI of

0.877–0.996 for PLS-DA, AUC of 0.905 and 95% CI of 0.822–0.986 for random forest, AUC of 0.757 and 95% CI of 0.635–0.879 for decision tree, Appendix Table S4).

LASSO is widely used in scenarios with high-dimensional datasets, like metabolomics with small n (n as the sample number) and large p (p as the *m/z* feature number) (Xiao et al, 2022; Yao et al, 2023a). LASSO mitigated this risk of overfitting by applying an $L_1$-penalty to select the most relevant *m/z* features for classification, thus enhancing the robustness and generalizability of the constructed model (Tibshirani et al, 2010; Zou and Hastie, 2005). The optimized $L_1$-penalty of 0.6 in LASSO offered 81 *m/z* features for EC diagnosis with the best AUC of 0.957 (95% CI of 0.906–1.000) in the discovery cohort (Appendix Fig. S5B). Notably, we also confirmed that there was no overfitting of LASSO model, based on the permutation test ($p < 0.001$, Appendix Fig. S5C) and the consistent result in an independent validation cohort (AUC of 0.957 and 95% CI of 0.920–0.995, Fig. 4C), compared with the discovery cohort (AUC of 0.957 and 95% CI of 0.906–1.000).

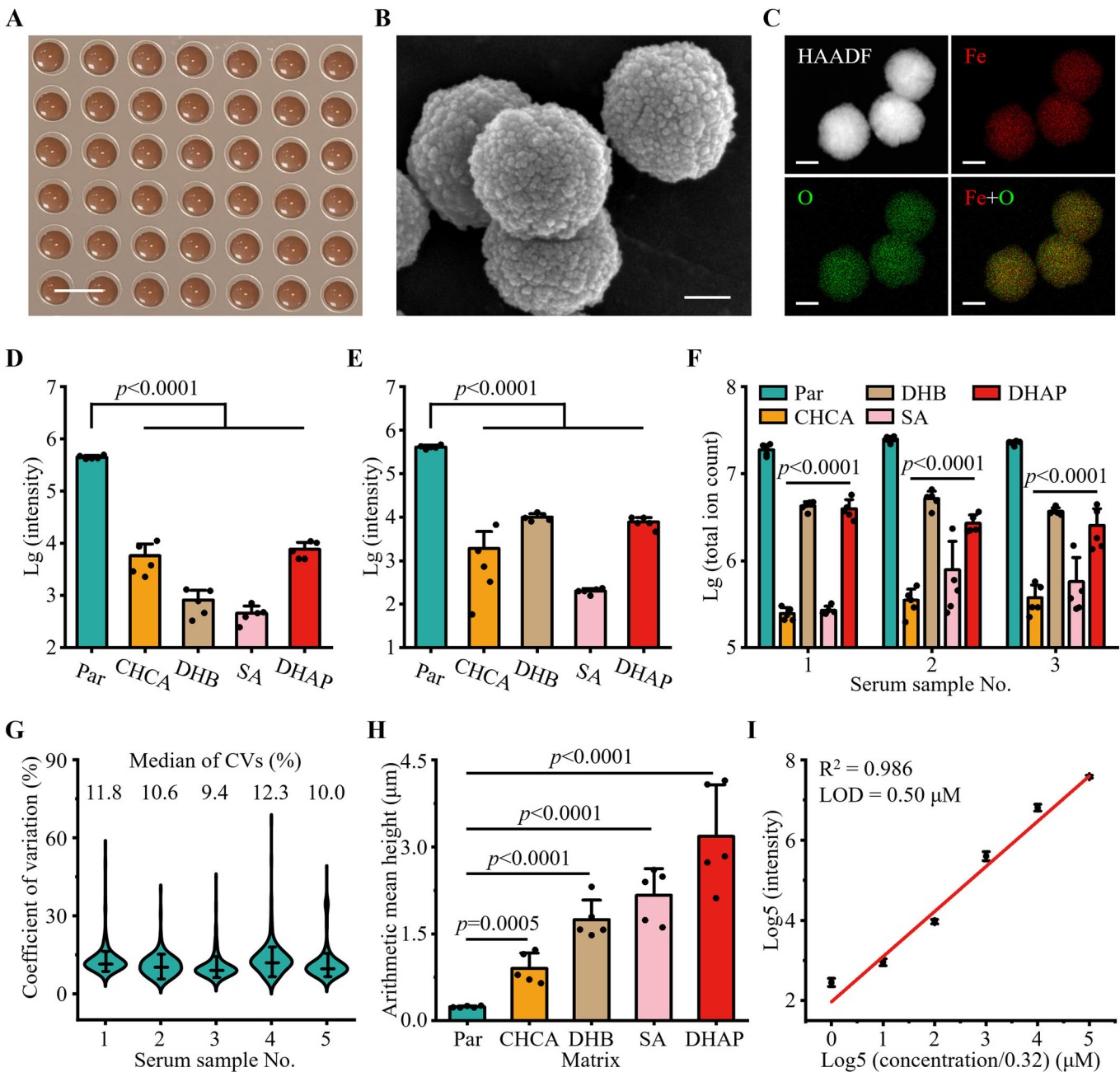

**Figure 2. High-performance detection of metabolites via PELDI-MS.**

(A) Digital image of the on-chip microarray showed the brown region after matrix printing. The scale bar was 5 mm. (B) Scanning electron microscopy image of the ferric oxide particles. The scale bar was 100 nm. (C) High-angle annular dark-field (HAADF) and elemental mapping analysis displayed the uniform distribution of Fe (in red) and O (in green) elements. The scale bar was 100 nm. (D, E) Sum of peak intensity of the standard sample under (D) high salt condition (20 mM of $Na^+$) and (E) biofluid-mimic condition (20 mM of $Na^+$ and 10 mg/mL of protein) for different matrices (ferric oxide particles (Par), α-cyano-4-hydroxycinnamic acid (CHCA), 2,5-dihydroxybenzoic acid (DHB), sinapic acid (SA), and 2,6-dihydroxyacetophenone (DHAP)). Data were mean ± SD, $N = 5$ technical replicates, two-tailed $t$-test. (F) Total ion count for serum samples using different matrices (Par, CHCA, DHB, SA, and DHAP). Data were mean ± SD, $N = 5$ technical replicates, two-tailed $t$-test. (G) Coefficient of variation (CV) distribution of intensities for $m/z$ features obtained from 5 serum samples in 10 independent technical replicates, demonstrating the high reproducibility (median CVs = 9.4–12.3%) of PELDI-MS. (H) Arithmetic mean height of co-crystallization morphology for various matrices (Par, CHCA, DHB, SA, and DHAP). Data were mean ± SD, $N = 5$ technical replicates, two-tailed $t$-test. (I) The PELDI-MS offered a good linear response ($R^2 = 0.986$) with a limit of detection (LOD) of 0.50 μM in proline analysis. Data were mean ± SD, $N = 3$ technical replicates. Source data are available online for this figure.

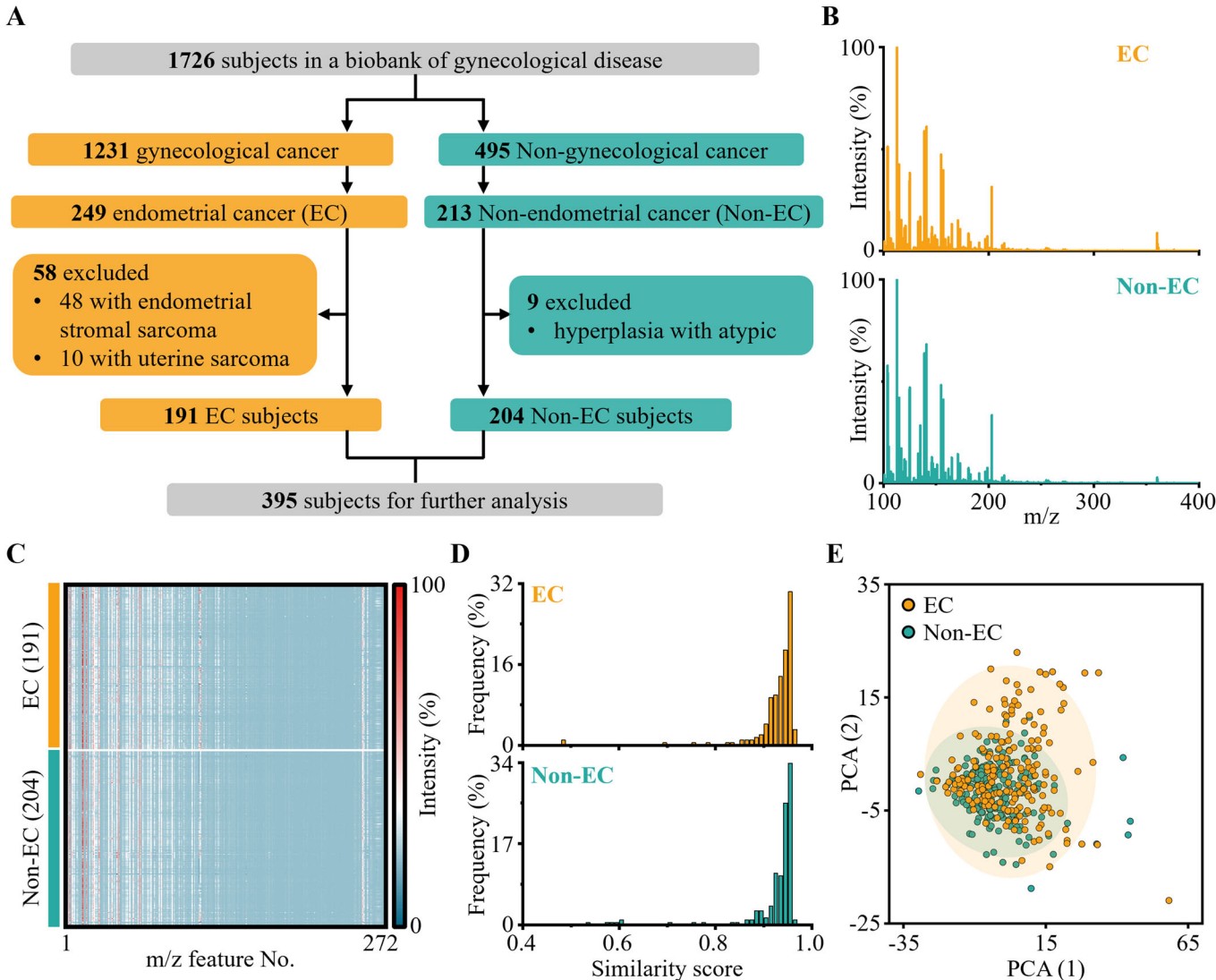

**Figure 3. Cohort design and SMFs characterization of EC and Non-EC.**

(A) The flow diagram showed the enrollment of EC and Non-EC subjects. Specifically, 395 subjects (191 EC and 204 Non-EC) were carefully chosen from 1726 subjects for further analysis. (B) Typical mass spectra of serum samples for EC and Non-EC subjects, displaying strong *m/z* features. (C) The blueprint of SMFs for 191 EC and 204 Non-EC serum samples was plotted using 272 *m/z* features. The color scale has been log-corrected. (D) Frequency distribution of the similarity score for EC and Non-EC groups was calculated to demonstrate the similarity of SMFs within the same group. (E) The unsupervised principal component analysis (PCA) showed a certain degree of overlap between EC and Non-EC groups. Source data are available online for this figure.

SMFs with LASSO model showed superior performance (AUC of 0.957, $p < 0.05$) for EC diagnosis in both discovery (Fig. 4B) and validation (Fig. 4C) cohorts, compared with the CA-125 (AUC of 0.670–0.684). Accordingly, SMFs with LASSO model afforded higher sensitivity (86.1–90.8%) with comparable specificity (91.4–91.9%) at the optimized Youden index than the CA-125 at the cut-off of 35 U/mL (sensitivity of 32.9–37.4% and specificity of 74.8–91.4%, Fig. 4D, Appendix Fig. S5D–F, and Appendix Table S5). Further, the enhancement of AUC (from 0.610–0.639 to 0.958–0.968, $p < 0.05$, Fig. 4E,F) and sensitivity (from 22.1–26.3% to 85.7–91.2%, Fig. 4G and Appendix Table S5) was also observed for early-stage (FIGO 2018 stage I/II) EC. Notably, for four typical cases (two cases of early-stage EC and two cases of Non-EC), the outcomes were correctly predicted using SMFs with LASSO model

(probability cut-off of 0.50), consistent with the clinical diagnosis (Fig. 4H). Thus, the machine learning of SMFs allowed for EC diagnosis even at an early stage, with better performance than the CA-125.

## Identification of biomarker panel for EC diagnosis

We identified a metabolic biomarker panel from SMFs for EC diagnosis toward potential large-scale clinical use. Specifically, 7 *m/z* features were initially obtained from SMFs at the mean intensity ($\bar{I}$) ≥ 100, AUC ≥ 0.70, and LASSO score ≥ 0.01 (Fig. 5A). Subsequently, we measured accurate *m/z* of the 7 *m/z* features at high resolution (<3 ppm) and confirmed 3 molecular formulas corresponding to 6 *m/z* features (Appendix Table S6). Finally,

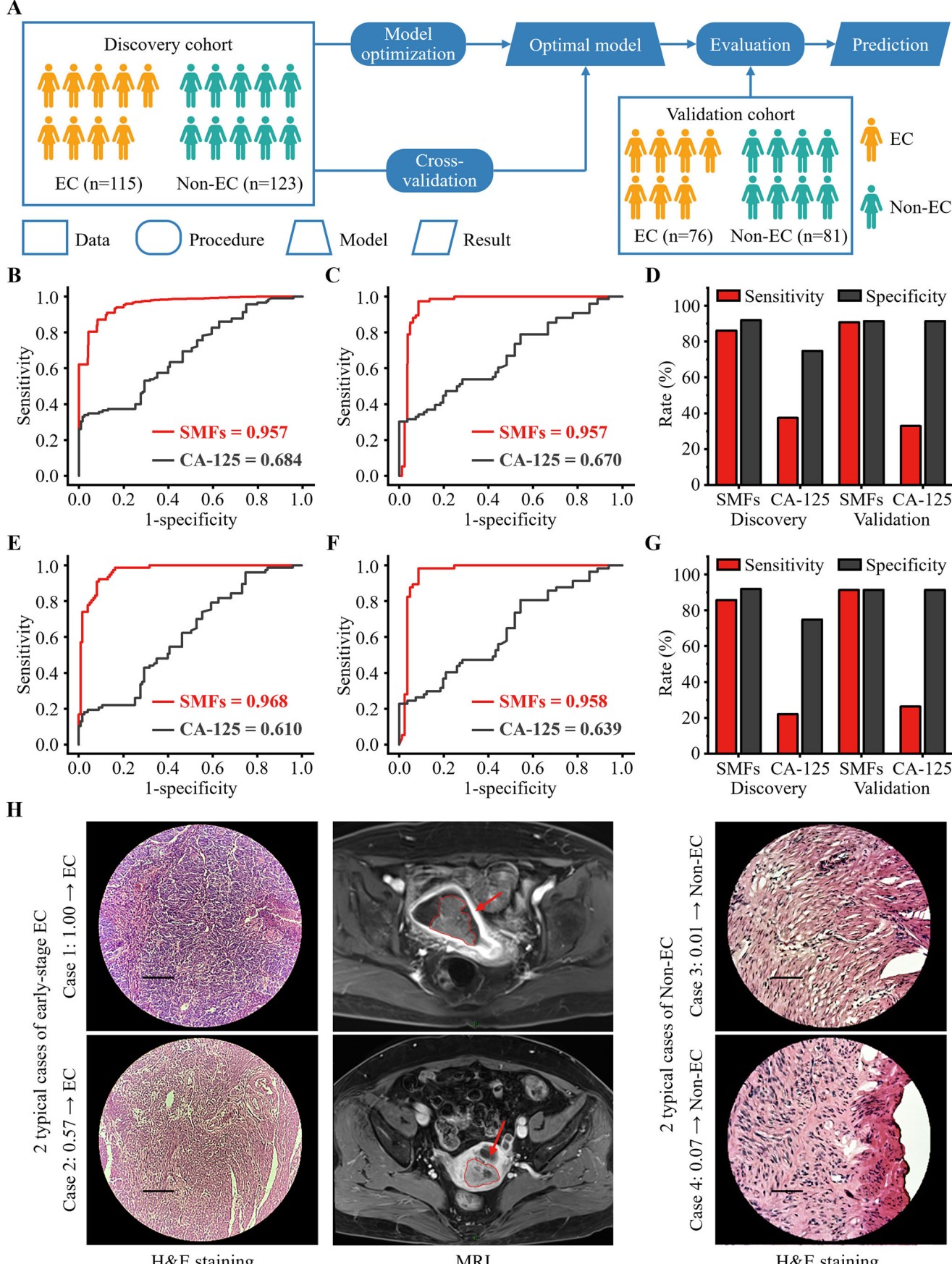

**Figure 4. Machine learning of SMFs for EC diagnosis.**

(A) Study design for EC diagnosis using machine learning of SMFs. 238 subjects (115/123, EC/Non-EC) were included in the discovery cohort to perform optimization. The optimal model was evaluated in an independent validation cohort (76/81, EC/Non-EC). (B, C) The receiver operator characteristic (ROC) curves for SMFs with least absolute shrinkage and selection operator (LASSO) model and cancer antigen 125 (CA-125) to distinguish EC and Non-EC in the (B) discovery cohort and (C) independent validation cohort. (D) The comparison of sensitivity between SMFs with LASSO model and CA-125 for distinguishing EC and Non-EC in both the discovery and validation cohorts. (E, F) The ROC curves for SMFs with LASSO model and CA-125 to distinguish early-stage (FIGO 2018 stage I/II) EC and Non-EC in the (E) discovery cohort and (F) independent validation cohort. (G) The comparison of sensitivity between SMFs with LASSO model and CA-125 for distinguishing early-stage EC and Non-EC in both the discovery and validation cohorts. (H) The 4 typical cases (2 cases of early-stage EC and 2 cases of Non-EC) were correctly predicted using SMFs with LASSO model (probability cut-off of 0.50), consistent with the clinical diagnosis (hematoxylin-eosin staining (H&E staining, 20×) and magnetic resonance imaging (MRI)). The scale bars were 100 μm and the tumors in MRI scans were marked with red arrows. Source data are available online for this figure.

based on the molecular formula, we identified a biomarker panel of three potential metabolites (including glutamine, glucose, and cholesterol linoleate) by searching the human metabolome database (Appendix Table S6). Notably, the clustering analysis of the 3 metabolites showed an accuracy (defined as the ratio of the number of accurately predicted samples to the total number of samples) of ~80.0% in distinguishing EC and Non-EC groups, confirming the ability of the constructed metabolic biomarker panel (Fig. 5B).

Specifically, glutamine and glucose were up-regulated ($p < 0.05$), while cholesterol linoleate was down-regulated ($p < 0.05$) in EC subjects (Fig. 5C). Further, we applied ultra-performance LC-MS (UPLC-MS) to validate the constructed metabolic biomarker panel in 60 subjects (30/30, EC/Non-EC, randomly selected from the 191 EC and 204 Non-EC subjects). The chromatographic comparison with standard metabolites and low CVs (4.3–10.7%) in quality control samples confirmed the reliability of UPLC-MS validation (Fig. 5D; Appendix Fig. S6A–C). Specifically, the upregulation of glutamine and glucose and the down-regulation of cholesterol linoleate were also observed in UPLC-MS ($p < 0.05$, Fig. 5E), demonstrating the robustness of the constructed metabolic biomarker panel.

Notably, the univariate receiver operator characteristic curve (ROC) analysis of a single metabolic biomarker showed an AUC of 0.697–0.738 and an accuracy of 69.0–75.4% in both EC (Fig. 5F) and early-stage EC (Fig. 5G) diagnosis. In comparison, the Met-score (probability in logistic regression of glutamine, glucose, and cholesterol linoleate) distinguished EC from Non-EC subjects with an AUC of 0.901–0.902 and an accuracy of 82.8–83.1% (Fig. 5F,G), demonstrating the superiority of the biomarker panel for EC diagnosis. To evaluate the effect of age, BMI, diabetes, and menopause on the identified biomarker panel, we computed the odds ratio of the Met-score and potentially relevant covariates (age, BMI, diabetes, and menopause). As a result, BMI, diabetes, and menopause were not significant covariates ($p > 0.05$) for the biomarker panel, while age was a significant covariate ($p < 0.05$) in the EC diagnosis (Appendix Table S7). This finding highlights the need to consider age in the interpretation and application of our biomarker panel. Notably, the older patients (average Met-score = 0.736) showed a slightly higher but not significant ($p > 0.05$) Met-score than the younger patients (average Met-score = 0.716), demonstrating the universal diagnostic performance of Met-score for different age groups. We achieved an AUC of 0.917–0.928 with an accuracy of 83.7–84.8% for EC diagnosis by combining the Met-score and CA-125 using a logistic regression (Fig. 5H). Notably, we observed a significantly ($p < 0.05$) increased diagnostic score (defined as the probability of being diagnosed as EC for the combined analysis) in advanced EC subjects (stage III/IV, average

score of 0.85), compared with early EC subjects (stage I/II, average score of 0.75). Thus, we confirmed the capability of the constructed metabolic biomarker panel for EC diagnosis.

## Biological function validation of metabolite biomarkers

To validate the biological function of metabolite biomarkers, 2 typical EC cell lines (ECC1 and Ishikawa) were used in an array of in vitro experiments (e.g., proliferation, colony formation, migration, and apoptosis). Initially, we confirmed the influence of glutamine, glucose, and cholesterol linoleate on the proliferation of EC cells (ECC1 and Ishikawa, Fig. 6A). Glucose promoted cell proliferation significantly at concentrations ranging from 5–20 mM, while cholesterol linoleate inhibited EC cell proliferation in a concentration-dependent manner ($p < 0.05$). In addition, glutamine slightly inhibited the proliferation of ECC1 cells at concentrations of 10–20 mM ($p < 0.05$), but showed no discernible effect at concentrations of >1 mM in Ishikawa cells ($p > 0.05$). Notably, consistent results were also observed in the clonal formation tests of EC cells (Fig. 6B).

Subsequently, we conducted the scratch assay on EC cell lines to characterize the cellular migration ability (Fig. 6C). The results showed that only glucose promoted wound healing in ECC1 cells at concentrations of 10 mM and 20 mM ($p < 0.05$), while there was no effect on the wound healing process in Ishikawa cells ($p > 0.05$). In contrast, neither glutamine nor cholesterol linoleate significantly influenced the wound healing process ($p > 0.05$), indicating their limited role in cellular migration. Further, we employed flow cytometric analysis to examine the cell apoptosis in different concentrations of metabolites (Appendix Fig. S7). We discovered that glutamine and glucose did not significantly impact apoptosis ($p > 0.05$, Appendix Fig. S7A, B). Intriguingly, cholesterol linoleate initiated apoptosis in the Ishikawa cells at a concentration of 300 μM (Appendix Fig. S7C). Altogether, our results validated the effect of glutamine, glucose, and cholesterol linoleate on various cellular biological processes, including proliferation, colony formation, cell migration, and apoptosis in EC.

## Discussion

EC remains a globally significant gynecological malignancy, with ~420,000 new cases diagnosed annually, highlighting the critical need for early diagnosis of EC to improve patient outcomes in clinical practice (Crosbie et al, 2022; O'Flynn et al, 2021). The transvaginal ultrasound is the initial investigation for abnormal symptoms (e.g., abnormal uterine bleeding) or physical

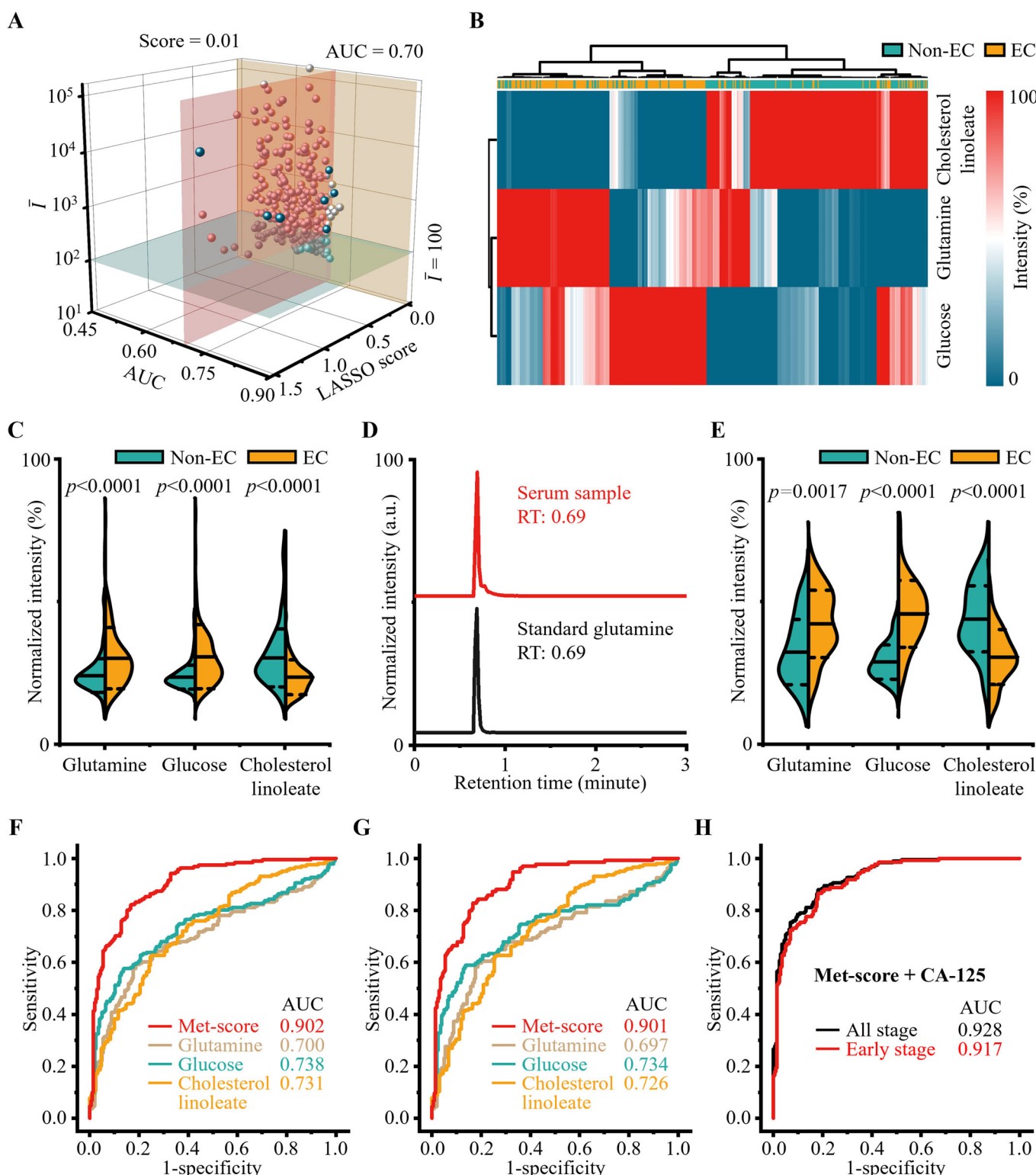

examination for gynecological diseases in clinics, which can identify the diseased region (e.g., endometrium, cervix uteri, and ovary). However, it is limited in EC diagnosis with a specificity of ~51.5% at the endometrial thickness cut-off of 5 mm (Jones et al, 2021). Biopsy and curettage are the gold standard for EC diagnosis.

However, the endometrial biopsy or curettage is an invasive procedure with potential risks, including pain, bleeding, and infection (O'Flynn et al, 2021). Further, the clinically used blood biomarker of CA-125 has a sensitivity of 32.9–37.4% for EC diagnosis. It is also impractical for early diagnosis with a sensitivity

◄

**Figure 5. Identification of biomarker panel for EC diagnosis.**

(A) 7 *m/z* features were obtained from the SMFs using the criteria of mean intensity (Ī) ≥ 100, area-under-the-curve (AUC) ≥ 0.70, and LASSO score ≥ 0.01. (B) Clustering analysis of the metabolic biomarkers of glutamine, glucose, and cholesterol linoleate demonstrated excellent separation, with an accuracy of 80.0% in differentiating EC from Non-EC. (C) The violin plot displayed the differential expression of the three metabolites detected by PELDI-MS in EC (sample number, N = 191) and Non-EC (sample number, N = 204). Metabolite levels were presented as normalized intensities. Data were mean ± SD, two-tailed *t*-test. (D) Chromatograms of standard glutamine and serum samples confirmed the reliability of glutamine detection in UPLC-MS validation. (E) The violin plot displayed the differential expression of the three metabolites detected by UPLC-MS in EC (sample number, N = 30) and Non-EC (sample number, N = 30). Metabolite levels were presented as normalized intensities. Data were mean ± SD, two-tailed *t*-test. (F, G) ROC curve analysis revealed a higher AUC of 0.901–0.902 for the metabolic biomarker panel (Met-score) in (F) EC and (G) early-stage EC diagnosis, compared to a single metabolic biomarker (AUC of 0.697–0.738). (H) ROC curve analysis demonstrated an AUC of 0.928 and 0.917 for EC and early-stage EC diagnosis by combining the Met-score and CA-125. Source data are available online for this figure.

of 22.1–26.3%, owing to its relatively normal level in early-stage EC (Jones et al, 2021; Njoku et al, 2019). Consequently, there is a growing interest in developing alternative biomarkers in the blood to enable timely differentiation diagnosis of EC and Non-EC, toward potential large-scale clinical use.

Several blood-based molecular biomarkers have been developed for EC diagnosis, focusing on genes, proteins, and metabolites. Regarding genes and proteins, the upstream molecules in the pathways, several biomarkers (*e.g.*, cell-free DNA and CA-125) are developed for EC diagnosis and showed an AUC of ~0.60–0.90 with biochemical reaction-based signal amplification assay (Cicchillitti et al, 2017; Jia et al, 2013; Knific et al, 2017; Li et al, 2018; Martinez-Garcia et al, 2018; Torres et al, 2013). In parallel, metabolite biomarkers provide a direct snapshot of the disease phenotype (Buergel et al, 2022; Yao et al, 2023b; Zhang et al, 2023). Moreover, metabolic reprogramming is recognized as a hallmark of malignancy and the reprogrammed metabolic activities can be exploited for diagnostic purposes (Faubert et al, 2020; Lopez-Otin et al, 2023). However, the current metabolite biomarkers in blood for EC diagnosis remain limited, given the traditional analytical techniques and lack of validated biological function (Ihata et al, 2014; Knific et al, 2018; Troisi et al, 2018; Yan et al, 2022). In this work, we successfully screened three metabolic biomarkers and achieved a desirable diagnostic performance (an AUC of >0.9 and an accuracy of >80.0%) with a reaction-free signal enhancement strategy, owing to the advanced analytical techniques and robust biomarker validations with a designed EC cohort.

For analytical techniques, the PELDI-MS showed the advantages of simple sample treatment, fast analytical speed, and low test cost compared with the other typical MS techniques like LC/GC-MS. For simple sample treatment and fast analytical speed, the high salt and protein tolerance of the tailored particles in PELDI-MS allowed direct detection of metabolites in serum (with 60–80 mg/mL of proteins and 135–145 mM of Na$^+$) free of sample treatment, therefore afford a fast analytical speed of ~30 s per sample. The LC/GC-MS require deproteinization and LC/GC in sample treatment to purify and enrich metabolites, with an analytical speed of ~30–60 min (Cao et al, 2020; Chen et al, 2022; Sato et al, 2022). For low test cost, the test cost of the PELDI-MS was ~3 dollars, considering all consumables (e.g., ferric oxide particles and standard metabolites for calibration) and equipment depreciation (e.g., laser generator and mass detector) (Chen et al, 2023). In comparison, the costs of LC/GC MS were usually ~tens of dollars, with the additional reagents (e.g., reagents for deproteinization) and instruments (e.g., chromatographic instrument) for sample treatment (Sato et al, 2022). Further, the on-chip microarray design in NELDI-MS allowed the automatic detection of metabolites with

high reproducibility (CVs of 5.6–11.0%), facile for potential large-scale tests in clinics.

Further, cohort design is fundamental in developing high-performance biomarker panels, providing precise insights into disease phenotypes. Most previous studies for EC diagnosis were conducted with a cohort size of n = 40–326 (Ihata et al, 2014; Knific et al, 2018; Troisi et al, 2018; Yan et al, 2022). In parallel, our study filtered 395 subjects of EC and Non-EC from 1726 subjects within a gynecological disease biobank, applying well-defined inclusion and exclusion criteria. Further, the power analysis of 12 samples (6/6, EC/Non-EC) confirmed a sample number of 200 (100/100, EC/Non-EC) achieved a predicted power of 0.85, indicating the statistical confidence level of the machine learning results (Li et al, 2019; Xia and Wishart, 2011). As a result, we successfully constructed a reliable clinical cohort for subsequent analysis.

For robust biomarker validations, the upregulation of glutamine and glucose and the downregulation of cholesterol linoleate were validated in UPLC-MS. Further, our results indicated that glucose facilitated cell proliferation, while glutamine exhibited an inhibitory effect on EC cell proliferation. Consistently, the importance of glucose was widely reported in tumor cells, which rely on glycolysis to proliferate and metastasize (Reinfeld et al, 2021; Zappasodi et al, 2021). Moreover, previous research has elucidated that glutamine inhibits tumor growth in melanoma, and dietary supplementation enhances survival in melanoma models (Ishak Gabra et al, 2020). Interestingly, while substantial research has been conducted on cholesterol and linoleic acid in cancer, there is a lack of literature regarding cholesterol linoleate (Huang et al, 2020a; Nava Lauson et al, 2023). Our study showed that cholesterol linoleate impedes cell proliferation by inducing apoptosis in EC cell lines. Collectively, these results provided biological insights for its use as diagnostic biomarkers.

In summary, we recorded the high-performance SMFs of 395 subjects and achieved an AUC of 0.957–0.968 in the EC diagnosis, outperforming the CA-125. Further, we identified a metabolic biomarker panel of three metabolites and achieved an AUC of 0.917–0.928 by combining with CA-125. Finally, we validated the effect of the three metabolites on EC cell behaviors in vitro. Therefore, our work would facilitate the development of novel diagnostic biomarkers for EC in clinics. Regarding the limitations, the following aspects should be mentioned: (1) mass spectrometer and designed particles are crucial for obtaining high-performance SMFs, and further engineering may be required for point-of-care tests. (2) this study focuses on the single-center retrospective population, and further research is required for the prospective population from multicenter to demonstrate the relevance and applicability of our findings to clinical settings. (3)

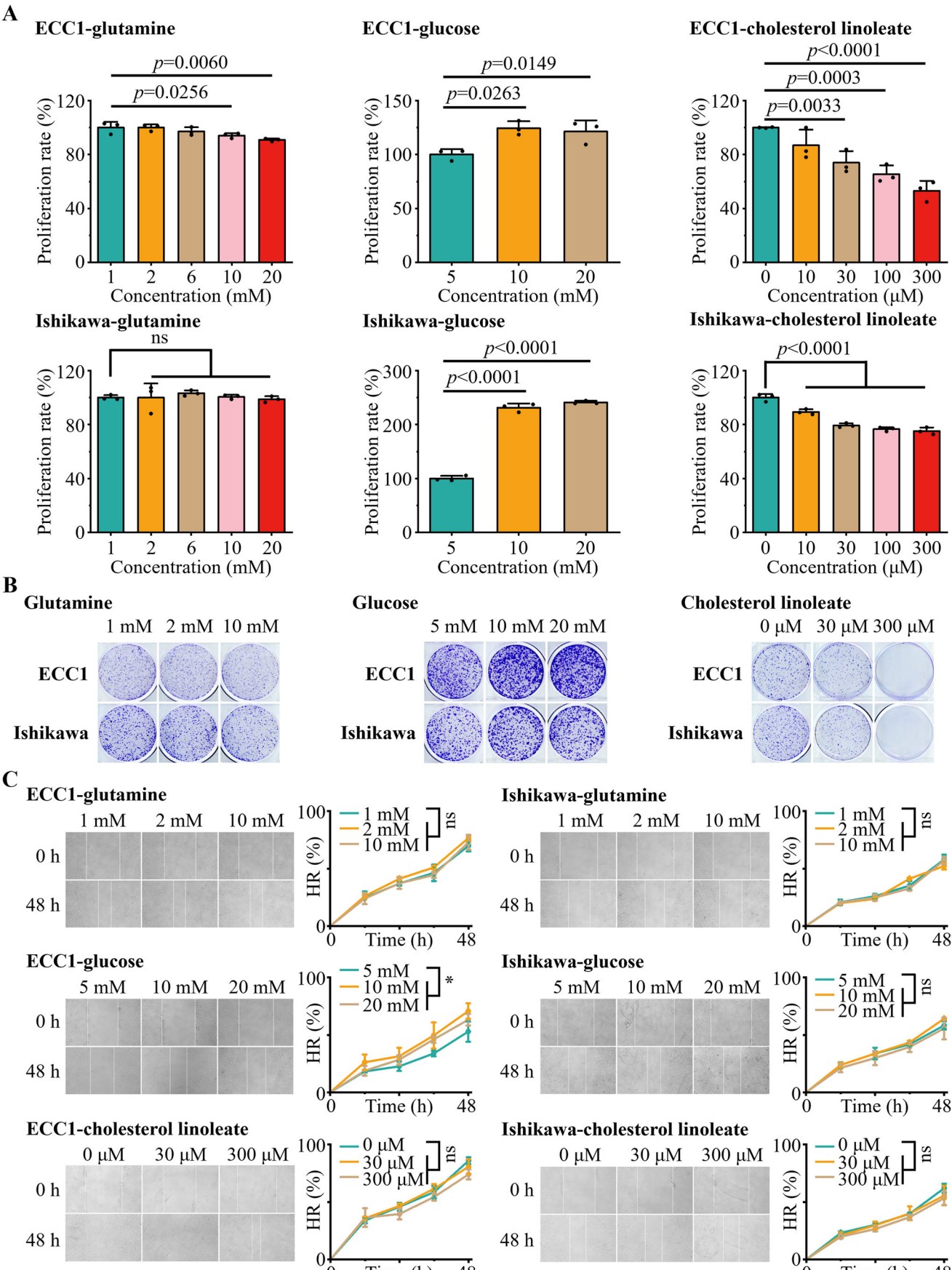

Figure 6. Biological function validation of metabolite biomarkers.

(A) The impact of glutamine, glucose, and cholesterol linoleate on the proliferation of EC cells. A slight inhibition was observed only in ECC1 cells for glutamine at concentrations of 10 mM and 20 mM. Glucose was found to promote proliferation in both EC cell lines, while cholesterol linoleate exhibited suppressive activity. Data were mean ± SD, $N = 3$ biological replicates, one-way ANOVA, ns (no significance, $p > 0.05$). (B) The impact of glutamine, glucose, and cholesterol linoleate on the colony formation of EC cells. The clonogenic assay was employed to verify cell colony formation ability. (C) Cell migration ability was estimated using a scratch wound healing assay. Glucose promoted wound healing for ECC1 cells at concentrations above 5 mM ($p = 0.0297$ for 10 mM and $p = 0.0487$ for 20 mM) but exhibited no effect on Ishikawa cells. Glutamine and cholesterol linoleate were observed to have no impact on wound healing in the scratch assay. Data were mean ± SD, $N = 3$ biological replicates, one-way ANOVA, ns (no significance, $p > 0.05$), *$p < 0.05$, HR (healing rate). Source data are available online for this figure.

further biological function validation of cholesterol linoleate and related pathways is needed. (4) The biomarker panel construction for differentiating different gynecological cancer types is also critical and needs further study.

# Methods

## Chemicals and reagents

This work included reagents for preparation of matrices (organic matrices and particles), standard metabolites, and other reagents. For organic matrices, trifluoroacetic acid (TFA, $C_2HF_3O_2$, 99.5%) was obtained from Macklin Biochemical Co., Ltd. (Shanghai, China). The α-cyano-4-hydroxycinnamic acid (CHCA, $C_{10}H_7NO_3$, 99.0%), 2,6-dihydroxyacetophenone (DHAP, $C_8H_8O_3$, 99.5%), sinapic acid (SA, $C_{11}H_{12}O_5$, 99.0%), and 2,5-dihydroxybenzoic acid (DHB, $C_7H_6O_4$, 99.0%) were purchased from Sigma-Aldrich (St. Louis, MO, USA). For particles, the trisodium citrate dihydrate ($Na_3C_6H_5O_7 \cdot 2H_2O$, 99.5%), ethylene glycol ($C_2H_6O_2$, 99.5%), and ferric chloride hexahydrate ($FeCl_3 \cdot 6H_2O$, 99.0%) were purchased from Sinopharm Chemical Reagent (Shanghai, China). The sodium acetate anhydrous ($NaC_2H_3O_2$, 99.0%) was procured from Aladdin Reagent (Shanghai, China). For standard metabolites, the thyroxine ($C_{15}H_{11}I_4NO_4$, 98.0%) was acquired from Innochem Technology Co., Ltd. (Beijing, China). The maltotriose ($C_{18}H_{32}O_{16}$, 98.0%) and lactose ($C_{12}H_{22}O_{11}$, 99.0%) were purchased from Sinopharm Chemical Reagent (Shanghai, China). The alanine ($C_3H_7NO_2$, 98.5%), proline ($C_5H_9NO_2$, 99.0%), glutamic acid ($C_5H_9NO_4$, 99.0%), glutamine ($C_5H_{10}N_2O_3$, 99.0%), cholesterol linoleate ($C_{45}H_{76}O_2$, 98.0%), and glucose ($C_6H_{12}O_6$, 99.5%) were purchased from Sigma-Aldrich (St. Louis, MO, USA). For other reagents, the methanol ($CH_4O$, HPLC), acetonitrile ($C_2H_3N$, HPLC), isopropanol ($C_3H_8O$, UPLC), dichloromethane ($CH_2Cl_2$, HPLC), and chloroform ($CHCl_3$, 99.0%) were purchased from Sigma-Aldrich (St. Louis, MO, USA). All chemical reagents were directly used without further purification. Deionized water (DIW, 18.2 MΩ·cm, Milli-Q, Millipore GmbH, Tuttlingen, Germany) was used in this work.

## Synthesis and characterization of particles

The ferric oxide particles were synthesized using a modified solvothermal approach (Huang et al, 2020b). Specifically, 0.6 g of ferric chloride hexahydrate and 0.15 g of trisodium citrate dihydrate were dissolved in 25 mL of ethylene glycol. Subsequently, 0.96 g of anhydrous sodium acetate was added, and the resultant mixture underwent a solvothermal reaction. Finally, the ferric oxide particles were washed and dried for further use. For characterization of the particles, 3D confocal reconstruction images were

obtained using a KEYENCE VK-X3000 (KEYENCE Co., Ltd., Osaka, Japan). Digital image of the microarray was captured using the P40 Pro (Huawei Technologies Co., Ltd., China). Dynamic light scattering (DLS) and zeta potential measurements were performed using a Mastersizer 2000 instrument (Malvern Instruments Inc., Worcestershire, UK). The ultraviolet–visible (UV–Vis) was obtained using a UV-3600 spectrophotometer (Shimadzu Ltd., Tokyo, Japan). Scanning electron microscopy was performed with a Gemini 300 (ZEISS Ltd., Oberkochen, Germany). Transmission electron microscopy (TEM), selected area electron diffraction (SAED), high-resolution TEM (HRTEM), and elemental mapping were conducted using a JEOL JEM-2100F (JEOL Ltd., Akishima, Japan).

## Clinical subjects and serum sample collection

This study adhered strictly to the principles of the Declaration of Helsinki and the Department of Health and Human Services Belmont Report, with the approval by the ethics committee of Renji Hospital, School of Medicine, Shanghai Jiao Tong University (2018-114). Before the study, written informed consent was obtained from all subjects. Serum samples from 1726 subjects were collected in a gynecological disease biobank at Renji Hospital, School of Medicine, Shanghai Jiao Tong University. Among these, 395 serum samples of 191 endometrial cancer (EC) and 204 Non-EC subjects were included according to pre-established criteria described in Fig. 3A, and no sample or data points were omitted in further analysis. The following data were collected: (1) clinical characteristics (e.g., age at pathological diagnosis, level of cancer antigen 125 (CA-125), body mass index (BMI), diabetes, meno-pause, hypertension, FIGO 2018 stage, and histology type) and (2) hematoxylin-eosin staining (H&E staining) and magnetic reso-nance imaging (MRI). Blood was collected through venipuncture and then centrifuged at $2000 \times g$ for 10 min. The serum was transferred to a microtube immediately and stored at −80 °C. The samples of the EC and Non-EC groups were collected during a similar period from Dec. 2018 to Sep. 2021, and the serum metabolic fingerprints (SMFs) database was recorded in Jul. 2022 using the serum samples that underwent one freeze-thaw cycle. The pathologists were blinded to any information about SMFs analysis.

## LDI-MS experiments

The LDI-MS experiments were conducted on Autoflex (time-of-flight mass spectrometer (TOF-MS), Bruker) or SolariX 7.0 T (Fourier transform-ion cyclotron resonance mass spectrometer (FT-ICR-MS), Bruker). 10 mg of organic matrices were dissolved in 1 mL of TA30 solution (acetonitrile:0.1% TFA solution, 3:7, v/v),

and the particle powder was dispersed in DIW with 1.0 mg/mL. 1.5 μL of either standard metabolite solutions or serum samples (10-fold dilution) were spotted on a 384 polished steel plate and dried. The serum samples were prepared randomly to minimize subjective bias for the SMFs database construction. Subsequently, matrix solution was added and dried before detection. Mass-to-charge ratio (*m/z*) calibration was performed using alanine, proline, glutamic acid, glucose, lactose, maltotriose, and thyroxine. For TOF-MS, the pulse frequency and laser shots per analysis were set to 1000 Hz and 2000, respectively. For FT-ICR-MS, the voltage of the detector plate was maintained at 210 V and the plate offset was set to 80 V.

## Sample treatment for UPLC-MS

The serum samples were prepared and analyzed randomly for metabolite and lipid extraction. For metabolite extraction (Liang et al, 2020), 80 μL of serum sample was initially added to 400 μL of extraction solution (methanol:acetonitrile, 1:1, v/v) and incubated at −20 °C to facilitate protein precipitation. The sample was centrifuged at 12,000 rpm for 15 min, and the supernatant was collected and evaporated to dryness. The dried extract was reconstituted in 140 μL of methanol:water (3:7, v/v) solution. For lipid extraction (Song et al, 2020), 50 μL of serum sample was added to 600 μL of methanol:chloroform (1:2, v/v) solution. The mixture was vortexed and then incubated for 1 h at 4 °C. After incubation, 170 μL of ice-cold DIW was added and placed at 4 °C for 10 min. The sample was centrifuged, and the lower organic phase was transferred into a new tube. Then, 400 μL of cleaning solution (chloroform:methanol:water, 85:14:1, v/v/v) was added to the remaining aqueous/methanol phase. The mixture was vortexed for 30 s and incubated for 30 min at 4 °C. The sample was then placed for 10 min and centrifuged at 12,000 rpm for 10 min. The lower organic phase from this step was combined with the organic extract obtained from the first round. The pooled organic extract was evaporated to dryness, and the dried extract was reconstituted in 140 μL of dichloromethane:isopropanol:methanol (1:1:2, v/v/v) solution, followed by thorough vortexing. A quality control (QC) sample was pooled with serum samples from 15 EC and 15 Non-EC subjects for metabolite and lipid extraction.

## UPLC-MS experiments

Metabolite and lipid extracts were analyzed using a Q Exactive Plus Mass Spectrometer (ThermoFisher Scientific, Waltham, MA, USA) connected to a Vanquish UPLC system (ThermoFisher Scientific, Waltham, MA, USA). For metabolite detection, reversed-phase LC (RPLC) was carried out using an ACQUITY UPLC HSS T3 column (100 × 2.1 mm, 1.7 μm, Waters). The mobile phase solvents consisted of 0.1% formic acid in water (A) and 0.1% formic acid in acetonitrile (B). For lipid detection, RPLC experiments used an ACQUITY UPLC BEH C18 column (100 × 2.1 mm, 1.7 μm, Waters). The mobile phase solvents were composed of 10 mM ammonium formate and 0.1% formic acid in acetonitrile/water (6:4, v/v) (A), as well as 10 mM ammonium formate and 0.1% formic acid in isopropanol/acetonitrile (9:1, v/v) (B). The flow rate was set at 0.4 mL/min, and the injection volume was 1 μL.

For both metabolite and lipid detection, the capillary temperature was maintained at 320 °C. In metabolite detection, the spray voltage was set to 3.2 kV for the positive mode and 2.8 kV for the negative mode. In lipid detection, the spray voltage was adjusted to 3.8 kV for the positive mode and 3.0 kV for the negative mode. Data were recorded at a full scan resolution of 70,000 using Xcalibur 3.0 software (ThermoFisher Scientific, Waltham, MA, USA). No smoothing procedures were applied, and spectra were used directly.

## Cell culture and reagents

The human EC cell lines, ECC1 and Ishikawa, were obtained from the American Type Culture Collection (Manassas, VA, USA) and authenticated via short tandem repeat (STR) profiling. Cells were cultured in RPMI-1640 (BasalMedia, Shanghai, China) supplemented with 2% FBS (ExCell Bio, Shanghai, China), 100 U/ml penicillin, and 0.1 mg/ml streptomycin (Cienry, Huzhou, China). Glucose studies were conducted by adding varying concentrations of glucose into glucose-free RPMI-1640 medium (BasalMedia, Shanghai, China) supplemented with 2% FBS. Cell cultures were maintained in a standard tissue culture incubator at 37 °C, with a relative humidity of 95% and an atmosphere of 5% carbon dioxide. All cell lines were mycoplasma negative, as the Share-bio PCR Mycoplasma Test Kit (Share-bio, Shanghai, China) determined. CCK-8 Kit and 488A-Annexin V/PI Apoptosis Kit were purchased from Share-bio (Shanghai, China). The 4% fixative solution was purchased from ABCONE (Shanghai, China). Crystal violet powder was purchased from Sangon Biotech (Shanghai, China).

## Biological function validation

Biological function validation of metabolite biomarkers in this work included CCK-8 assay, colony formation assay, wound healing assay, and apoptosis assay. The CCK-8 assay followed the provided protocol for the CCK-8 Kit. Briefly, an initial population of 4000 cells was cultured in each well of a 96-well plate overnight. Subsequently, the cells were treated with various concentrations of glutamine, glucose, and cholesterol linoleate. After a cultivation period of 96 h, the CCK-8 working liquid was added to each well and incubated for 1 h before analysis. The optical density (OD) at 490 nm was obtained using a plate reader. The cell proliferation rate was evaluated by comparing the OD values of treated cells to that of an untreated control group. For the colony formation assay, ECC1 and Ishikawa cells were seeded in 6-well plates (1000 cells per well) and treated with glutamine, glucose, or cholesterol linoleate, respectively. Cell clones were fixed with 4% fixative solution and stained with 0.1% crystal violet. The stained plates were scanned using a desktop scanner (Epson Co., Ltd., Beijing, China).

For the wound healing assay, $1 \times 10^6$ ECC1 and Ishikawa cells per well were seeded for 18 h before scratched with a 200 μL plastic pipette tip to generate a cell-free area. Then, it was washed with phosphate-buffered saline (PBS) 3 times, and different concentrations of glutamine, glucose, and cholesterol linoleate were added. Photos were taken at 0 h and every 12 h for 48 h at the same positions using an inverted microscope (ZEISS Ltd., Oberkochen, Germany). The healing rate was calculated using the formula: healing rate = (initial wound area − non-healing area) / initial wound area × 100%. The Apoptosis assay used the 488A-Annexin V/PI Apoptosis Kit. ECC1 and Ishikawa cells were seeded at a

density of $2 \times 10^5$ cells per well and treated with glutamine, glucose, and cholesterol linoleate for 72 h. The cells were then harvested using 0.25% trypsin (without ethylenediaminetetraacetic acid), washed twice with PBS, and re-suspended in 100 μL of annexin-binding buffer. The cells were stained with 5 μL of Annexin V and 4 μL of PI at 4 °C for 20 min. Following staining, each sample was added with 400 μL of PBS and analyzed using a BD LSRFortessa flow cytometer (BD Biosciences, NJ, USA). Apoptosis profiles were determined by Flowjo software. All experiments were performed 3 times.

## Data processing

Data processing included data processing in LDI-MS, UPLC-MS, and machine learning using Python version 3.8. For data processing in LDI-MS, the baseline correction and spectral smoothing were first carried out to eliminate noise in raw mass spectra. Subsequently, the *m/z* features with a S/N ≥ 3 were extracted using the scipy package in peak detection (Virtanen et al, 2020). Finally, peak alignment and filtration were performed, retaining only *m/z* features present in ≥2/3 of the samples for EC/Non-EC groups (Liu et al, 2022; Yu et al, 2006). The adduct species and molecular formula of *m/z* features were putatively annotated in the Bruker Compass DataAnalysis software (version 5.0) based on the accurate *m/z* measurements in FT-ICR-MS (Sun et al, 2021). In addition, the metabolites were identified through the molecular formula search against the Human Metabolome Database (HMDB, https://hmdb.ca/) (Wishart et al, 2022). Notably, the sum of intensities of the multiple adduct forms was used for each metabolite in the analysis. The LOD was calculated using the following formula: LOD = 3 σ/S (Hu et al, 2023; Kim et al, 2023). Where σ is the standard deviation of the method and S is the slope of the calibration curve. For data processing in UPLC-MS, the raw LC-MS data files were imported into the TraceFinder 5.0 General Quan software (ThermoFisher Scientific, Waltham, MA, USA) for peak detection and identification. For peak detection, parameters such as area and peak noise factors were configured at 5 and 10, respectively. Furthermore, the baseline window and minimum peak width were optimized to 40 and 3, ensuring the efficient processing of the data. An S/N threshold of 3 was established to identify peaks. The *m/z* features of the SMFs database were standardized to meet the assumption of normal distribution, and the variation of each *m/z* feature within and between the groups was evaluated and statistically compared. For machine learning, the LASSO, logistic regression, PLS-DA, random forest, and decision tree, were trained to predict the clinical outcomes of EC patients using the scikit-learn package (Pedregosa et al, 2012). The machine learning algorithms were trained using 5-fold cross-validation 20 times in the discovery cohort, and evaluated in an independent validation cohort.

## Statistical analysis

Principal component analysis (PCA), t-distributed stochastic neighbor embedding (t-SNE), uniform manifold approximation and projection (UMAP), and permutation test were performed on Python version 3.8, using scikit-learn and umap-learn packages (Mcinnes and Healy, 2018; Pedregosa et al, 2012). Power analysis was conducted on MetaboAnalyst 5.0 (https://www.metaboanalyst.ca/) to determine the sample

number required for statistically significant machine learning (Pang et al, 2021). Specifically, the SMFs of 12 serum samples (6/6, EC/Non-EC) were uploaded, and the predicted power was computed at a false discovery rate (FDR) of 0.1. The adjusted cosine similarity algorithm was employed to calculate the similarity score. Statistical analysis in this study included Student's *t*-test, analysis of variance (ANOVA), and Delong test. In particular, *p* values for the Student's *t*-test were calculated to compare LDI-MS performance, co-crystallization of matrices, clinical characteristics, and expression of metabolites using Microsoft Excel (Office 2019). The ANOVA was performed for all the experiments in biological function validation using GraphPad Prism software (version 8.0.2, GraphPad Software Inc., USA). The Delong test was performed to compare the performance of SMFs with CA-125 using Rstudio 2022.12.0. In this study, the significance level was set at 0.05 for all analysis.

## Data availability

The mass spectrometry data from this publication have been deposited to the NIH Common Fund's National Metabolomics Data Repository (NMDR) Website (https://www.metabolomicsworkbench.org), the Metabolomics Workbench (Sud et al, 2016), and assigned the identifier of ST003048.

## Peer review information

---

### The paper explained

**Problem**

Endometrial cancer (EC) is the most prevalent gynecological cancer worldwide. Existing diagnostic tools, including transvaginal ultrasound, biopsy, and curettage, are limited in terms of specificity (~51.1% for transvaginal ultrasound) and invasiveness (endometrial sampling for biopsy and curettage). Moreover, the commonly used blood biomarker for EC diagnosis, cancer antigen 125 (CA-125), exhibits low sensitivity (<60%).

**Results**

We recorded the high-performance serum metabolic fingerprints (SMFs) of 191 EC and 204 Non-EC subjects via particle-enhanced laser desorption/ionization mass spectrometry (PELDI-MS). Further, we identified a metabolic biomarker panel of glutamine, glucose, and cholesterol linoleate with an AUC of 0.901–0.902 and an accuracy of 82.8–83.1% for EC diagnosis through machine learning of SMFs. Finally, we validated the function of the three metabolite biomarkers on EC cell behaviour, including proliferation, colony formation, migration, and apoptosis in vitro.

**Impact**

Our work will facilitate the development of novel diagnostic biomarkers for EC. Functional validation of these metabolite biomarkers provides biological insights for their use as diagnostic biomarkers.

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

## Acknowledgements

The authors gratefully acknowledge financial support from Projects YG2021GD02, YG2023ZD08, YG2024ZD07, TMSK-2021-207, TMSK-2021-124, NRCTM(SH)-2021-06 by Shanghai Jiao Tong University School of Medicine, Projects 2021YFA0910100, 2021YFF0703500, 2022YFC2502800, 2022YFE0103500 by National Key R&D Program of China, Projects 82173077 and 82211540402 by NSFC, Project 20ZR1433100 by Shanghai Natural Science Foundation, and Project 2021-01-07-00-02-E00083 by Shanghai Institutions of Higher Learning. This work was also sponsored by the Innovation Group Project of the Shanghai Municipal Health Commission (2019CXJQ03), Innovation Research Plan by the Shanghai Municipal Education Commission (ZXWF082101), and the Science and Technology Commission of Shanghai Municipality (20DZ2220400). We thank all the participants for donating their bio-samples and clinical information to this study.

## Author contributions

**Wanshan Liu**: Data curation; Software; Formal analysis; Validation; Investigation; Visualization; Methodology; Writing—original draft; Writing—review and editing. **Jinglan Ma**: Data curation; Software; Formal analysis; Validation; Investigation; Visualization; Methodology; Writing—review and editing. **Juxiang Zhang**: Formal analysis; Validation; Investigation; Writing—review and editing. **Jing Cao**: Validation; Methodology; Writing—review and editing. **Xiaoxiao Hu**: Investigation; Methodology; Writing—review and editing. **Yida Huang**: Validation; Methodology; Writing—review and editing. **Ruimin Wang**: Investigation; Methodology; Writing—review and editing. **Jiao Wu**: Validation; Methodology; Writing—review and editing. **Wen Di**: Conceptualization; Resources; Supervision; Validation; Investigation; Project administration; Writing—review and editing. **Kun Qian**: Conceptualization; Resources; Supervision; Funding acquisition; Validation; Methodology; Project administration; Writing—review and editing. **Xia Yin**: Conceptualization; Resources; Supervision; Funding acquisition; Validation; Investigation; Visualization; Project administration; Writing—review and editing.

## Disclosure and competing interests statement

The authors declare the following competing interests. The authors have filed patents using the methods and technologies to analyze metabolites.

