## [Peer Review File · EMBO Molecular Medicine]

Identification and validation of serum metabolite biomarkers for endometrial cancer diagnosis

Wanshan Liu, Jinglan Ma, Juxiang Zhang, Jing Cao, Xiaoxiao Hu, Yida Huang, Ruimin Wang, Jiao Wu, Wen Di, Qian Kun, and Xia Yin

Corresponding authors: Qian Kun (k.qian@sjtu.edu.cn) , Wen Di (diwen@renji.com), Xia Yin (yinxia@renji.com)

Review Timeline:

Submission Date:	23rd Sep 23
Editorial Decision:	26th Oct 23
Revision Received:	2nd Dec 23
Editorial Decision:	19th Jan 24
Revision Received:	24th Jan 24
Accepted:	26th Jan 24

Editor: Lise Roth

Transaction Report:

26th Oct 2023

Dear Prof. Kun,

Thank you for the submission of your manuscript to EMBO Molecular Medicine, and please accept my apologies for the delay in getting back to you as we were waiting for one of the referees' reports. We have now received feedback from the two reviewers who agreed to evaluate your manuscript. As you will see below, the reviewers raise substantial concerns on your work, which unfortunately preclude its publication in EMM in its current form.

The reviewers find that the question addressed by the study is of potential interest, however they also raise a number of serious concerns. If you feel you can satisfactorily address the points listed by the referees, you may wish to submit a revised version of your manuscript. Please attach a covering letter giving details of the way in which you have handled each of the points raised by the referees. A revised manuscript will once again be subject to review, and we cannot guarantee at this stage that the eventual outcome will be favorable.

We are expecting your revised manuscript within three months, if you anticipate any delay, please contact us.

We require:

- 1) A .docx formatted version of the manuscript text (including legends for main figures, EV figures and tables). Please make sure that the changes are highlighted to be clearly visible.
- 2) Individual production quality figure files as .eps, .tif, .jpg (one file per figure). For guidance, download the 'Figure Guide PDF' (<https://www.embopress.org/page/journal/17574684/authorguide#figureformat>).
- 3) At EMBO Press we ask authors to provide source data for the main figures. Our source data coordinator will contact you to discuss which figure panels we would need source data for and will also provide you with helpful tips on how to upload and organize the files.
- 4) A .docx formatted letter INCLUDING the reviewers' reports and your detailed point-by-point responses to their comments. As part of the EMBO Press transparent editorial process, the point-by-point response is part of the Review Process File (RPF), which will be published alongside your paper.
- 5) A complete author checklist, which you can download from our author guidelines (<https://www.embopress.org/page/journal/17574684/authorguide#submissionofrevisions>). Please insert information in the checklist that is also reflected in the manuscript. The completed author checklist will also be part of the RPF.
- 6) Please note that all corresponding authors are required to supply an ORCID ID for their name upon submission of a revised manuscript.
- 7) It is mandatory to include a 'Data Availability' section after the Materials and Methods. Before submitting your revision, primary datasets produced in this study need to be deposited in an appropriate public database, and the accession numbers and database listed under 'Data Availability'. Please remember to provide a reviewer password if the datasets are not yet public (see <https://www.embopress.org/page/journal/17574684/authorguide#dataavailability>). In case you have no data that requires deposition in a public database, please state so in this section. Note that the Data Availability Section is restricted to new primary data that are part of this study.
- 8) For data quantification: please specify the name of the statistical test used to generate error bars and P values, the number (n) of independent experiments (specify technical or biological replicates) underlying each data point and the test used to calculate p-values in each figure legend. The figure legends should contain a basic description of n, P and the test applied. Graphs must include a description of the bars and the error bars (s.d., s.e.m.). Please provide exact p values.
- 9) Our journal encourages inclusion of *data citations in the reference list* to directly cite datasets that were re-used and obtained from public databases. Data citations in the article text are distinct from normal bibliographical citations and should directly link to the database records from which the data can be accessed. In the main text, data citations are formatted as follows: "Data ref: Smith et al, 2001" or "Data ref: NCBI Sequence Read Archive PRJNA342805, 2017". In the Reference list,

data citations must be labeled with "[DATASET]". A data reference must provide the database name, accession number/identifiers and a resolvable link to the landing page from which the data can be accessed at the end of the reference. Further instructions are available at .

13) Author contributions: CRediT has replaced the traditional author contributions section because it offers a systematic machine readable author contributions format that allows for more effective research assessment. Please remove the Authors Contributions from the manuscript and use the free text boxes beneath each contributing author's name in our system to add specific details on the author's contribution. More information is available in our guide to authors.

16) As part of the EMBO Publications transparent editorial process initiative (see our Editorial at <http://embomolmed.embopress.org/content/2/9/329>), EMBO Molecular Medicine will publish online a Review Process File (RPF) to accompany accepted manuscripts.

In the event of acceptance, this file will be published in conjunction with your paper and will include the anonymous referee reports, your point-by-point response and all pertinent correspondence relating to the manuscript. Let us know whether you agree with the publication of the RPF and as here, if you want to remove or not any figures from it prior to publication.

I look forward to receiving your revised manuscript.

Yours sincerely,

Lise Roth

***** Reviewer's comments *****

Referee #1 (Remarks for Author):

In this work, the authors report the identification and validation of a panel of 3 metabolites capable of discriminating endometrial cancer (EC) and Non-EC with high accuracy. The paper is well organized and represents an impressive interdisciplinary work. The MS technique seems very promising at bringing metabolomics profiling to the clinic, showing a good linear response and capable of high throughput. The identified biomarker panel accurately discriminates between endometrial cancer (EC) and Non-EC. However, I believe some aspects of the work require more attention. Please find a point-by-point list below of the aspects that should be improved in this paper:

-
Major points:

1. The author claimed that PELDI-MS offered a good linear response ($R^2 = 0.963-0.986$) in metabolite detection. The related LOD of the 3 typical metabolites should also be provided.
2. How does the author define accuracy in this paper? Please specify in the paper. Further, the accuracy for combining the Met-score, menopause, and CA-125 is missing.
3. The author achieved an AUC of 0.938-0.944 for EC diagnosis by combining the Met-score, menopause, and CA-125. What model is used for the combined analysis, and is there any difference for EC patients of early and advanced stages in the combined analysis.
4. There is no reference to other interference sources in the manuscript, such as the storage time, storage condition, and freeze/thaw cycle of the serum sample. This is crucial in metabolic analysis.
5. More details on data processing in LDI-MS for getting the 272 m/z features should be provided so the reader can follow the work more efficiently.
6. The statistical tests used for each comparison and the related replicate number should be described more clearly.

Minor points:

7. Throughout the manuscript, abbreviations used in figures and tables should be spelled out in their legends (e.g., Figure S2A-B).
8. The average size and zeta potential of the nanoparticles should be provided in Figure S1.
9. As multiple adducts were detected, is the analysis including only one or multiple adduct forms?
10. What is the cutoff of CA-125 used in this work and clinics?

Referee #2 (Remarks for Author):

Introduction: This paper presents the outcomes of a noteworthy investigation into the use of serum metabolic profiling for the study of Endometrial Cancer (EC), emphasizing the biological validation of a panel comprising three selected metabolites. The study employs an innovative mass spectrometry-based technique for metabolomics profiling and involves a relatively large patient cohort (n=191 EC + 204 Non-EC subjects). While the research holds value and reveals intriguing insights, several substantial concerns and minor issues necessitate resolution before recommending it for publication.

Major Concerns:

1. Lack of Panel Specificity: The selection of three metabolites, namely glutamine, glucose, and cholesterol linoleate, for inclusion in the panel is problematic. These metabolites have been associated with metabolic perturbations in various cancers, making it imperative to evaluate the panel's specificity concerning EC. The Warburg effect and glutamine addiction are common metabolic reprogramming features in many cancer types, not limited to EC. Furthermore, the authors derived EC cases from a cohort encompassing diverse gynecological diseases, both oncological and non-oncological. The absence of data on other cancer types raises concerns about the panel's specificity.

2. Impact of Comorbidities on Metabolite Concentrations: EC risk factors such as obesity, diabetes, hypertension, hypercholesterolemia, hypertriglyceridemia, and hyperuricemia significantly influence metabolite concentrations, including carnitines, amino acids, and sugars. Table S3 reveals significant differences ($p < 0.001$) in age, BMI, and diabetes prevalence between EC and non-EC patients. These disparities could introduce bias during the training phase and panel development. The potential impact of these differences on the results remains unaddressed. Moreover, the preponderance of post-menopausal EC cases compared to non-menopausal non-EC cases could also influence the training phase, warranting a comprehensive assessment of these differences.

3. Hystotype-Specific Findings: As the majority of EC cases are of the endometrioid histotype (88.5%), it is essential to specify that the metabolomics profiling results and the applicability of the panel are primarily relevant to this specific histotype.

Additionally, although the biological validation of selected biomarkers is commendable, the true litmus test for such a proposal lies in clinical validation, which is absent in this paper.

Minor Concerns:

1. Applicability of PELDI-MS: The choice of PELDI-MS, a technique suitable for analyzing samples with limited volume availability, appears unjustified for serum samples. This decision lacks appropriate rationale and necessitates clarification, given the instrument costs and the relatively low sensitivity for several metabolites.

2. Choice of Classification Model: The paper deviates from the common practice of employing Partial Least Squares Discriminant Analysis (PLS-DA), which is widely used in metabolomics, opting instead for various regression-based models for classification. The reasoning behind this unconventional choice should be provided for enhanced clarity.

3. Addressing Model Overfitting: The paper overlooks the potential issue of model overfitting, given the higher number of analyzed features in comparison to the sample size. To mitigate this risk, introducing a feature selection strategy, such as the Boruta algorithm or genetic algorithm, could prove useful. Furthermore, it is essential to conduct an overfitting estimation, such as a permutation test, to account for the imbalance between features and observations.

4. Age Discrepancy Reporting: The paper presents an inconsistency in age reporting. While Table S3 indicates a significant age difference ($p < 0.001$) between the studied groups, the Results section suggests that the mean age of cases and controls in the training group does not differ ($p > 0.05$). This inconsistency requires clarification and explanation.

In conclusion, this paper offers promise in exploring the potential of serum metabolic profiling in EC research. However, addressing the major concerns pertaining to panel specificity, the influence of comorbidities, and the hystotype-specific nature of findings, along with resolving the minor issues, will significantly enhance the paper's quality and its suitability for publication in a reputable scientific journal.

Responses to the reviewers' comments

Reviewer 1:

Comment: In this work, the authors report the identification and validation of a panel of 3 metabolites capable of discriminating endometrial cancer (EC) and Non-EC with high accuracy. The paper is well organized and represents an impressive interdisciplinary work. The MS technique seems very promising at bringing metabolomics profiling to the clinic, showing a good linear response and capable of high throughput. The identified biomarker panel accurately discriminates between endometrial cancer (EC) and Non-EC. However, I believe some aspects of the work require more attention. Please find a point-by-point list below of the aspects that should be improved in this paper.

Response:

We thank the reviewer's appreciation of the importance of our work and the insightful comments. We thoroughly revised our manuscript with specific changes highlighted in yellow to address all the points raised by the reviewer.

Comment 1: The author claimed that PELDI-MS offered a good linear response ($R^2 = 0.963-0.986$) in metabolite detection. The related LOD of the 3 typical metabolites should also be provided.

Response:

We thank the reviewer for the comments. The limit of detection (LOD) for the 3 typical metabolites was calculated as 0.41-0.53 μM .

Specifically, the LOD was calculated using the following formula: $\text{LOD} = 3 \sigma/S$ (Hu *et al*, 2023; Kim *et al*, 2023). Where σ is the standard deviation of the method and S is the slope of the calibration curve. As a result, the high reproducibility of PELDI-MS offered a good linear response ($R^2 = 0.963-0.986$) with a LOD of 0.41-0.53 μM in metabolite analysis (Fig R1).

We have added the related results and clarified the calculation of LOD in the manuscript and appendix as follows (Fig 2I and Appendix Fig S3F and G).

Figure R1. The PELDI-MS offered a good linear response ($R^2 = 0.963-0.986$) with a limit of detection (LOD) of 0.41-0.53 μM in (A) proline, (B) alanine, and (C) glucose analysis.

Responses:

"Notably, the high reproducibility of PELDI-MS offered a good linear response ($R^2 = 0.963-0.986$) with a limit of detection (LOD) of 0.41-0.53 μM in metabolite analysis (Fig 2I and Appendix Fig S3F and G)." (Page 5, Manuscript)

"The limit of detection (LOD) was calculated using the following formula: $\text{LOD} = 3 \sigma/S$ (Hu *et al*, 2023; Kim *et al*, 2023). Where σ is the standard deviation of the method and S is the slope of the calibration curve." (Page 20, Manuscript)

"I. The PELDI-MS offered a good linear response ($R^2 = 0.986$) with a limit of detection (LOD) of 0.50 μM in proline analysis. Data were mean \pm SD, $N = 3$ technical replicates." (Page 30, Manuscript, Figure 2I)

"F, G. The PELDI-MS offered a good linear response ($R^2 = 0.963-0.973$) with a limit of detection (LOD) of 0.41-0.53 μM in (F) alanine and (G) glucose analysis. Data were mean \pm SD, $N = 3$ technical replicates." (Page 4, Appendix, Appendix Figure S3F and G)

Comment 2: How does the author define accuracy in this paper? Please specify in the paper. Further, the accuracy for combining the Met-score, menopause, and CA-125 is missing.

Response:

We thank the reviewer for the insightful comments. The accuracy in this paper was defined as the ratio of the number of accurately predicted samples to the total number of samples. Further, the accuracy for the combined analysis was 83.7-84.8%.

We have clarified the definition of accuracy and added the accuracy for the combined analysis in the manuscript as follows.

Revisions:

"Notably, the clustering analysis of the 3 metabolites showed an accuracy (defined as the ratio of the number of accurately predicted samples to the total number of samples) of 80.0% in distinguishing EC and Non-EC groups, confirming the ability of the constructed metabolic biomarker panel (Fig 5B)." (Page 9, Manuscript)

"We achieved an AUC of 0.917-0.928 with an accuracy of 83.7-84.8% for EC diagnosis by combining the Met-score and CA-125 using a logistic regression (Fig 5H)." (Page 10, Manuscript)

Comment 3: The author achieved an AUC of 0.938-0.944 for EC diagnosis by combining the Met-score, menopause, and CA-125. What model is used for the combined analysis, and is there any difference for EC patients of early and advanced stages in the combined analysis?

Response:

We thank the reviewer for the thoughtful consideration. The logistic regression was used for the combined analysis. We observed a significantly ($P < 0.05$) increased diagnostic score (defined as the probability of being diagnosed as EC for the combined analysis) in advanced EC subjects (stage III/IV, average score of 0.85), compared with early EC subjects (stage I/II, average score of 0.75).

We have added the related results and clarifications in the manuscript as follows.

Revisions:

"We achieved an AUC of 0.917-0.928 with an accuracy of 83.7-84.8% for EC diagnosis by combining the Met-score and CA-125 using a logistic regression (Fig 5H). Notably, we observed a significantly ($P < 0.05$) increased diagnostic score (defined as the probability of being diagnosed as EC for the combined analysis) in advanced EC subjects (stage III/IV, average score of 0.85), compared with early EC subjects (stage I/II, average score of 0.75)." (Page 10, Manuscript)

Comment 4: There is no reference to other interference sources in the manuscript, such as the storage time, storage condition, and freeze/thaw cycle of the serum sample. This is crucial in metabolic analysis.

Response:

We thank the reviewer for pointing out this issue. The serum samples were stored at -80°C for three and a half years and underwent one freeze-thaw cycle for the serum metabolic fingerprints (SMFs) database.

Specifically, the serum was transferred to a microtube immediately and stored at -80°C. The samples of the EC and Non-EC groups were collected during a similar period from Dec. 2018 to Sep. 2021, and the SMFs database was recorded in Jul. 2022 using the serum samples that underwent one freeze-thaw cycle. The pathologists were blinded to any information about SMFs analysis.

We have added the related information in the manuscript as follows.

Revisions:

"The serum was transferred to a microtube immediately and stored at -80°C. The samples of the EC and Non-EC groups were collected during a similar period from Dec. 2018 to Sep. 2021, and the serum metabolic fingerprints (SMFs) database was recorded in Jul. 2022 using the serum samples that underwent one freeze-thaw cycle. The pathologists were blinded to any information about SMFs analysis." (Page 16, Manuscript)

Comment 5: More details on data processing in LDI-MS for getting the 272 m/z features should be provided so the reader can follow the work more efficiently.

Response:

We thank the reviewer for the thoughtful consideration. The data processing in LDI-MS for getting the 272 m/z features included baseline correction, spectral smoothing, peak detection, peak alignment, and peak filtration.

Specifically, the baseline correction and spectral smoothing were first carried out to eliminate noise in raw mass spectra. Subsequently, the m/z features with a signal-to-noise ratio (S/N) ≥ 3 were extracted using the scipy package in peak detection (Virtanen *et al*, 2020). Finally, peak alignment and filtration were performed, retaining only m/z

features present in $\geq 2/3$ of the samples for EC/Non-EC groups (Liu *et al*, 2022; Yu *et al*, 2006).

We have added the related information in the manuscript as follows.

Revisions:

"For data processing in LDI-MS, the baseline correction and spectral smoothing were first carried out to eliminate noise in raw mass spectra. Subsequently, the m/z features with a signal-to-noise ratio (S/N) ≥ 3 were extracted using the scipy package in peak detection (Virtanen *et al*, 2020). Finally, peak alignment and filtration were performed, retaining only m/z features present in $\geq 2/3$ of the samples for EC/Non-EC groups (Liu *et al*, 2022; Yu *et al*, 2006)." (Page 20, Manuscript)

Comment 6: The statistical tests used for each comparison and the related replicate number should be described more clearly.

Response:

We thank the reviewer for the comments. We have clarified the statistical test used for each comparison and the related replicate number in the manuscript and appendix as follows.

Revisions:

"Statistical analysis in this study included Student's *t*-test, analysis of variance (ANOVA), and Delong test. In particular, *P* values for the Student's *t*-test were calculated to compare LDI-MS performance, co-crystallization of matrices, clinical characteristics, and expression of metabolites using Microsoft Excel (Office 2019). The ANOVA was performed for all the experiments in biological function validation using GraphPad Prism software (version 8.0.2, GraphPad Software Inc., USA). The Delong test was performed to compare the performance of SMFs with CA-125 using Rstudio 2022.12.0. In this study, the significance level was set at 0.05 for all analysis." (Page 21, Manuscript)

"D, E. Sum of peak intensity of the standard sample under (D) high salt condition (20 mM of Na⁺) and (E) biofluid-mimic condition (20 mM of Na⁺ and 10 mg/mL of protein) for different matrices (ferric oxide particles (Par), α -cyano-4-hydroxycinnamic acid (CHCA), 2,5-dihydroxybenzoic acid (DHB), sinapic acid (SA), and 2,6-dihydroxyacetophenone (DHAP)). Data were mean \pm SD, *N* = 5 technical replicates,

two-tailed *t*-test, *****P* < 0.0001. F. Total ion count for serum samples using different matrices (Par, CHCA, DHB, SA, and DHAP). Data were mean ± SD, *N* = 5 technical replicates, two-tailed *t*-test, *****P* < 0.0001. H. Arithmetic mean height of co-crystallization morphology for various matrices (Par, CHCA, DHB, SA, and DHAP). Data were mean ± SD, *N* = 5 technical replicates, two-tailed *t*-test, ****P* < 0.001, *****P* < 0.0001." (Page 29, Manuscript, Figure 2D-F and Figure 2H)

"C. The violin plot displayed the differential expression of the 3 metabolites detected by PELDI-MS in EC (*N* = 191) and Non-EC (*N* = 204). Metabolite levels were presented as normalized intensities. Data were mean ± SD, two-tailed *t*-test, *****P* < 0.0001. E. The violin plot displayed the differential expression of the 3 metabolites detected by UPLC-MS in EC (*N* = 30) and Non-EC (*N* = 30). Metabolite levels were presented as normalized intensities. Data were mean ± SD, two-tailed *t*-test, ****P* < 0.001, *****P* < 0.0001." (Page 34, Manuscript, Figure 5C and Figure 5E)

"A. The impact of glutamine, glucose, and cholesterol linoleate on the proliferation of EC cells. A slight inhibition was observed only in ECC1 cells for glutamine at concentrations of 10 mM and 20 mM. Glucose was found to promote proliferation in both EC cell lines, while cholesterol linoleate exhibited suppressive activity. Cell proliferation was analyzed using the cell counting kit-8 (CCK-8) assay. Data were mean ± SD, *N* = 3 biological replicates, one-way ANOVA, ns (no significance), **P* < 0.05, ***P* < 0.01, ****P* < 0.001, *****P* < 0.0001. C. Cell migration ability was estimated using a scratch wound healing assay. Glucose promoted wound healing for ECC1 cells at concentrations above 5 mM but exhibited no effect on Ishikawa cells. Glutamine and cholesterol linoleate were observed to have no impact on wound healing in the scratch assay. Data were mean ± SD, *N* = 3 biological replicates, one-way ANOVA, ns (no significance), **P* < 0.05, HR (healing rate)." (Page 36, Manuscript, Figure 6A and Figure 6C)

"C. Cholesterol linoleate induced apoptosis in Ishikawa cells at 300 μM, while demonstrating no influence on ECC1 cells. Cell apoptosis was examined by flow cytometric analysis. Data were mean ± SD, *N* = 3 biological replicates, one-way ANOVA, ns (no significance), *****P* < 0.0001." (Page 8, Appendix, Appendix Figure S7)

Comment 7: Throughout the manuscript, abbreviations used in figures and tables should be spelled out in their legends (e.g., Figure S2A-B).

Response:

We thank the reviewer for pointing out this issue. We have now spelled out all the abbreviations used in figures and tables as follows.

Revisions:

"Figure 1. Schematics for biomarker panel identification and validation. A. We collected serum samples of 395 subjects (191 endometrial cancer (EC) and 204 Non-endometrial cancer (Non-EC)) and recorded the high-performance serum metabolic fingerprints (SMFs) via particle-enhanced laser desorption/ionization mass spectrometry (PELDI-MS) analysis. B. Then, we achieved EC diagnosis by machine learning of high-performance SMFs. C. Further, we identified a metabolic biomarker panel of glutamine, glucose, and cholesterol linoleate through accurate m/z and database search and validated in ultra-performance liquid chromatography-MS (UPLC-MS). D. Finally, we validated the effect of the 3 metabolite biomarkers on EC cell behaviors in vitro." (Page 28, Manuscript, Figure 1)

"G. Coefficient of variation (CV) distribution of intensities for m/z features obtained from 5 serum samples in 10 independent technical replicates, demonstrating the high reproducibility (median CVs = 9.4-12.3%) of PELDI-MS." (Page 30, Manuscript, Figure 2G)

"B, C. The receiver operator characteristic (ROC) curves for SMFs with least absolute shrinkage and selection operator (LASSO) model and cancer antigen 125 (CA-125) to distinguish EC and Non-EC in the (B) discovery cohort and (C) independent validation cohort." (Page 33, Manuscript, Figure 4B and C)

"A. The impact of glutamine, glucose, and cholesterol linoleate on the proliferation of EC cells. A slight inhibition was observed only in ECC1 cells for glutamine at concentrations of 10 mM and 20 mM. Glucose was found to promote proliferation in both EC cell lines, while cholesterol linoleate exhibited suppressive activity. Cell proliferation was analyzed using the cell counting kit-8 (CCK-8) assay. Data were mean \pm SD, $N = 3$ biological replicates, one-way ANOVA, ns (no significance), $*P < 0.05$, $**P < 0.01$, $***P < 0.001$, $****P < 0.0001$." (Page 36, Manuscript, Figure 6A)

"Appendix Figure S1. Characterization of the ferric oxide particles. A, B. (A) Transmission electron microscopy (TEM) image (scale bar = 100 nm) and (B) selected area electron diffraction (SAED) pattern (scale bar = 5 nm^{-1}) of the ferric oxide particles. C. High-resolution TEM (HRTEM) image displayed the crystal lattice of the ferric

oxide particles, marked by white circles. The scale bar was 5 nm. D. Ultraviolet-visible (UV-Vis) spectrum of the ferric oxide particles showed a strong absorbance at 355 nm. E, F. (E) Dynamic light scattering (DLS) and (F) zeta potential was recorded by 3 independent technical replicates. Data were mean \pm SD, $N = 3$ technical replicates." (Page 2, Appendix, Appendix Figure S1)

"Appendix Figure S2. High salt and protein tolerance of PELDI-MS. A, B. Typical mass spectrum of the standard sample under (A) high salt condition (20 mM Na⁺) and (B) biofluid-mimic condition (20 mM Na⁺ and 10 mg/mL protein) using ferric oxide particles. Alkali metal cation adduction ([M+Na]⁺, [M-H+2Na]⁺, and [M-2H+3Na]⁺) of small metabolites (alaline (Ala), proline (Pro), glutamic acid (Glu), glucose (Glc), and lactose (Lac)) was marked. C-F. Typical mass spectra of the standard sample under high salt condition (20 mM Na⁺) using (C) α -cyano-4-hydroxycinnamic acid (CHCA), (D) 2,5-dihydroxybenzoic acid (DHB), (E) sinapic acid (SA), and (F) 2,6-dihydroxyacetophenone (DHAP)." (Page 3, Appendix, Appendix Figure S2)

"B, C. The unsupervised analysis of (B) t-distributed stochastic neighbor embedding (t-SNE) and (C) uniform manifold approximation and projection (UMAP) of SMFs showed a certain degree of overlap between EC and Non-EC groups." (Page 5, Appendix, Appendix Figure S4B and C)

Comment 8: The average size and zeta potential of the nanoparticles should be provided in Figure S1.

Response:

We thank the reviewer for the comments. The average size and zeta potential of the nanoparticles were 252.5 ± 6.3 nm and -23.2 ± 0.5 mV, respectively (Fig R2).

Figure R2. (A) Dynamic light scattering (DLS) and (B) zeta potential were recorded by 3 independent technical replicates. Data were mean \pm SD, $N = 3$ technical replicates.

We have provided the average size and zeta potential of the nanoparticles in Appendix Fig S1.

Comment 9: As multiple adducts were detected, is the analysis including only one or multiple adduct forms?

Response:

We thank the reviewer for the thoughtful comments. The analysis included multiple adduct forms. Specifically, the sum of intensities of the multiple adduct forms was used for each metabolite in the analysis.

We have added the related information in the manuscript as follows.

Revisions:

"Notably, the sum of intensities of the multiple adduct forms was used for each metabolite in the analysis." (Page 20, Manuscript)

Comment 10: What is the cutoff of CA-125 used in this work and clinics?

Response:

We thank the reviewer for the comments. The 35 U/mL was used as the cut-off for CA-125 in this work and clinics (Ihata *et al*, 2014; Miyagi *et al*, 2016).

We have added the related information in the manuscript as follows.

Revisions:

"Accordingly, SMFs with LASSO model afforded higher sensitivity (86.1-90.8%) with comparable specificity (91.4-91.9%) at the optimized Youden index than the CA-125 at the cut-off of 35 U/mL (sensitivity of 32.9-37.4% and specificity of 74.8-91.4%, Fig 4D, Appendix Fig S5D-F, and Appendix Table S5)." (Page 8, Manuscript)

Reviewer 2:

Comment: This paper presents the outcomes of a noteworthy investigation into the use of serum metabolic profiling for the study of Endometrial Cancer (EC), emphasizing the biological validation of a panel comprising three selected metabolites. The study employs an innovative mass spectrometry-based technique for metabolomics profiling and involves a relatively large patient cohort (n=191 EC + 204 Non-EC subjects). While the research holds value and reveals intriguing insights, several substantial concerns and minor issues necessitate resolution before recommending it for publication.

Response:

We thank the reviewer for the kind consideration of this work. As described below in addressing all the points raised by the reviewer, we also thoroughly revised our manuscript with specific changes highlighted in yellow.

Comment 1: Lack of Panel Specificity: The selection of three metabolites, namely glutamine, glucose, and cholesterol linoleate, for inclusion in the panel is problematic. These metabolites have been associated with metabolic perturbations in various cancers, making it imperative to evaluate the panel's specificity concerning EC. The Warburg effect and glutamine addiction are common metabolic reprogramming features in many cancer types, not limited to EC. Furthermore, the authors derived EC cases from a cohort encompassing diverse gynecological diseases, both oncological and non-oncological. The absence of data on other cancer types raises concerns about the panel's specificity.

Response:

We thank the reviewer for pointing out this issue. The panel specificity can be addressed by the transvaginal ultrasound, which can identify the lesion region (*e.g.*, endometrium, cervix uteri, and ovary). The biomarker panel was used for differentiation diagnosis of EC and Non-EC in women with a determined lesion in the endometrium by ultrasound findings (*e.g.*, thickened endometrium or mass in the uterine cavity).

Specifically, transvaginal ultrasound is the initial investigation for abnormal symptoms (*e.g.*, abnormal uterine bleeding) or physical examination for gynecological diseases in clinics, which can identify the lesion region (*e.g.*, endometrium, cervix uteri, and ovary). Then, the biomarker panel in this work can be used for differentiation

diagnosis of EC and Non-EC in women with a determined lesion in the endometrium by ultrasound findings (*e.g.*, thickened endometrium or mass in the uterine cavity).

We have realized that the identified biomarkers (*e.g.*, glutamine and glucose) are associated with metabolic perturbations in various cancers, making differentiating various cancer types problematic. Further, while substantial research has been conducted on cholesterol and linoleic acid in cancer, there is a lack of literature regarding cholesterol linoleate, which can potentially be a biomarker specific to EC (Huang *et al*, 2020; Nava Lauson *et al*, 2023). The biomarker panel construction for differentiating different cancer types is also critical and will be a future direction in our work on gynecological diseases.

We have added the related clarifications and revised the manuscript as follows.

Revisions:

"Notably, differentiation diagnosis of abnormality detected by ultrasound findings (*e.g.*, thickened endometrium or mass in the uterine cavity) is essential and remains challenging in clinical practice. Herein, we identified a metabolic biomarker panel for differentiation diagnosis of EC using machine learning of high-performance serum metabolic fingerprints (SMFs) and validated the biological function." (Page 2, Manuscript)

"Therefore, timely differentiation diagnosis of EC and Non-EC in abnormality detected by ultrasound findings (*e.g.*, thickened endometrium or mass in the uterine cavity) is essential for optimal patient outcomes (Jones *et al*, 2021; Koskas *et al*, 2021). However, existing diagnostic tools, including transvaginal ultrasound, biopsy, and curettage, are limited by low specificity (~51.1% for transvaginal ultrasound at the endometrial thickness cut-off of 5 mm) or invasiveness (endometrial sampling for biopsy and curettage) (Jones *et al.*, 2021). Further, the commonly used blood biomarker for EC diagnosis in clinics is cancer antigen 125 (CA-125), limited by its low sensitivity (< 60%) (Njoku *et al*, 2019). Thus, there is an urgent need for alternative biomarkers in the blood to enable timely differentiation diagnosis of EC and Non-EC for potential clinical use." (Page 3, Manuscript)

"Herein, we identified a metabolic biomarker panel for differentiation diagnosis of EC using machine learning of high-performance SMFs and validated the biological function (Fig 1)." (Page 4, Manuscript)

"The transvaginal ultrasound is the initial investigation for abnormal symptoms (e.g., abnormal uterine bleeding) or physical examination for gynecological diseases in clinics, which can identify the diseased region (e.g., endometrium, cervix uteri, and ovary). However, it is limited in EC diagnosis with a specificity of ~51.5% at the endometrial thickness cut-off of 5 mm (Jones *et al.*, 2021)." (Page 11, Manuscript)

"4) The biomarker panel construction for differentiating different gynecological cancer types is also critical and needs further study." (Page 14, Manuscript)

Comment 2: Impact of Comorbidities on Metabolite Concentrations: EC risk factors such as obesity, diabetes, hypertension, hypercholesterolemia, hypertriglyceridemia, and hyperuricemia significantly influence metabolite concentrations, including carnitines, amino acids, and sugars. Table S3 reveals significant differences ($p < 0.001$) in age, BMI, and diabetes prevalence between EC and non-EC patients. These disparities could introduce bias during the training phase and panel development. The potential impact of these differences on the results remains unaddressed. Moreover, the preponderance of post-menopausal EC cases compared to non-menopausal non-EC cases could also influence the training phase, warranting a comprehensive assessment of these differences.

Response:

We thank the reviewer for the thoughtful comments. We have comprehensively evaluated the effect of age, BMI, hypertension, diabetes, and menopausal on the identified biomarker panel. We confirmed BMI, hypertension, diabetes, and menopausal would not introduce bias during the training phase and panel development. We apologize that the influence of hypercholesterolemia, hypertriglyceridemia, and hyperuricemia can not be evaluated due to the missing data in $> 95\%$ of patients. We believe that this additional analysis in the manuscript will provide a more thorough understanding of the impact of comorbidities on our findings and the applicability of our biomarker panel.

Specifically, the clinical characteristics of age, BMI, diabetes, menopause, and hypertension were collected and summarized in Table R1. The hypertension distribution showed no significant difference between the two groups ($P > 0.05$), while the age, BMI, diabetes, and menopause revealed significant differences ($P < 0.001$) between EC and non-EC groups. It is essential to mention that the imbalance is not intentional and reflects the natural prevalence of EC in the population.

Table R1. Clinical characteristics of the enrolled EC and Non-EC subjects.

Characteristic	EC ^{a)}	Non-EC ^{a)}	P value
Sample number	191	204	–
Age (year)			
Mean (range)	58 (27-81)	43 (17-79)	< 0.001
BMI			
Mean (range)	25 (17-46)	23 (15-34)	< 0.001
Diabetes			
(No/Yes/Unknown)	149/37/5	188/16/0	< 0.001
Menopause			
(No/Yes/Unknown)	61/125/5	164/40/0	< 0.001
Hypertension			
(No/Yes/Unknown)	131/56/4	158/46/0	0.11

To evaluate the effect of age, BMI, diabetes, and menopause on the identified biomarker panel, we computed the odds ratio of the Met-score (prediction score of the biomarker panel) and potentially relevant covariates (age, BMI, diabetes, and menopause). As a result, BMI, diabetes, and menopause were not significant covariates ($P > 0.05$) for the biomarker panel, while age was a significant covariate ($P < 0.05$) in the EC diagnosis (Table R2). This finding highlights the need to consider age in the interpretation and application of our biomarker panel. Notably, we have matched the age distribution ($P > 0.05$) between the EC and Non-EC groups in the discovery cohort to mitigate age bias during the training phase and panel development. As a result, the older patients (average Met-score = 0.736) showed a slightly higher but not significant ($P > 0.05$) Met-score than the younger patients (average Met-score = 0.716), demonstrating the universal diagnostic performance of Met-score for different age groups.

Table R2. Odds ratio of Met-score and potentially relevant variables (Age, BMI, diabetes, and menopause).

Covariate	Odds ratio (95% CI)	Significance (P value)
Met-score	7.356 (5.924-9.143)	< 0.001
Age	3.357 (2.465-4.572)	< 0.001
BMI	1.158 (0.973-1.379)	0.397
Diabetes	0.394 (0.240-0.645)	0.059
Menopause	1.241 (0.752-2.048)	0.666

We have added the related results in the manuscript and appendix as follows (Appendix Table S3 and Appendix Table S7).

Revisions:

"Clinical characteristics, like age at pathological diagnosis, level of CA-125, BMI, diabetes, menopause, hypertension, International Federation of Gynecology and Obstetrics (FIGO) 2018 stage, and histology type, are summarized (Appendix Table S3)." (Page 6, Manuscript)

"Notably, we have matched the age distribution ($P > 0.05$) between the EC and Non-EC groups in the discovery cohort to mitigate age bias during the training phase and panel development." (Page 7, Manuscript)

"To evaluate the effect of age, BMI, diabetes, and menopause on the identified biomarker panel, we computed the odds ratio of the Met-score and potentially relevant covariates (age, BMI, diabetes, and menopause). As a result, BMI, diabetes, and menopause were not significant covariates ($P > 0.05$) for the biomarker panel, while age was a significant covariate ($P < 0.05$) in the EC diagnosis (Appendix Table S7). This finding highlights the need to consider age in the interpretation and application of our biomarker panel. Notably, the older patients (average Met-score = 0.736) showed a slightly higher but not significant ($P > 0.05$) Met-score than the younger patients (average Met-score = 0.716), demonstrating the universal diagnostic performance of Met-score for different age groups." (Page 9, Manuscript)

Comment 3: Hystotype-Specific Findings: As the majority of EC cases are of the endometrioid histotype (88.5%), it is essential to specify that the metabolomics profiling results and the applicability of the panel are primarily relevant to this specific histotype. Additionally, although the biological validation of selected biomarkers is commendable, the true litmus test for such a proposal lies in clinical validation, which is absent in this paper.

Response:

We thank the reviewer for pointing out this issue. We have specified that our metabolomics profiling results and the applicability of the panel are primarily relevant to endometrioid histotypes of EC. Regarding the clinical validation of the identified biomarkers, we acknowledge the significance of clinical validation, and we will further validate the identified biomarkers on prospective populations from multicenter in the near future to confirm the relevance and applicability of our findings to clinical settings.

We have added the related clarifications in the manuscript as follows.

Revisions:

"The endometrioid is the main histotype in EC, accounting for ~80% of patients (Lu & Broaddus, 2020)." (Page 3, Manuscript)

"Notably, 88.5% of subjects in EC were endometrioid histotypes, indicating our further analysis was primarily relevant to this specific histotype." (Page 6, Manuscript)

"2) this study focuses on the single-center retrospective population, and further research is required for the prospective population from multicenter to demonstrate the relevance and applicability of our findings to clinical settings." (Page 14, Manuscript)

Comment 4: Applicability of PELDI-MS: The choice of PELDI-MS, a technique suitable for analyzing samples with limited volume availability, appears unjustified for serum samples. This decision lacks appropriate rationale and necessitates clarification, given the instrument costs and the relatively low sensitivity for several metabolites.

Response:

We thank the reviewer for the valuable comments. The rationales for the choice of PELDI-MS are simple sample treatment, fast analytical speed (~30 seconds per sample), and low test cost (~3 dollars). Further, the supplementary experiments showed that the PELDI-MS offered a sensitivity of 0.41-0.53 μM in metabolite analysis, sufficient to detect the identified biomarkers with concentrations of ~hundreds to thousands μM .

For simple sample treatment and fast analytical speed, the high salt and protein tolerance of the tailored particles in PELDI-MS allowed direct detection of metabolites in serum (with 60-80 mg/mL of proteins and 135-145 mM of Na^+) free of sample treatment, therefore afford a fast analytical speed of ~30 seconds per sample. In comparison, the other commonly used MS techniques like liquid/gas chromatography MS (LC/GC-MS) require deproteinization and LC/GC in sample treatment to purify and enrich metabolites, with an analytical speed of ~30-60 minutes (Cao *et al*, 2020; Chen *et al*, 2022; Sato *et al*, 2022).

For low test cost, the test cost of the PELDI-MS was ~3 dollars, considering all consumables (*e.g.*, ferric oxide particles and standard metabolites for calibration) and equipment depreciation (*e.g.*, laser generator and mass detector) (Chen *et al*, 2023). In comparison, the costs of LC/GC MS were usually ~tens of dollars, with the additional reagents (*e.g.*, reagents for deproteinization) and instruments (*e.g.*, chromatographic instrument) for sample treatment (Sato *et al*, 2022).

For sensitivity, we have added supplementary experiments to evaluate the sensitivity of PELDI-MS using 3 typical metabolites. As a result, the PELDI-MS offered a sensitivity of 0.41-0.53 μM in metabolite analysis (Fig R3), sufficient to detect the identified biomarkers with concentrations of ~hundreds to thousands μM .

Figure R3. The PELDI-MS offered a good linear response ($R^2 = 0.963$ - 0.986) with a limit of detection (LOD) of 0.41-0.53 μM in (A) proline, (B) alanine, and (C) glucose analysis.

We have added the related experimental details, results, and discussion in the manuscript and appendix as follows (Fig 2I and Appendix Fig S3F and G).

Revisions:

"Notably, the high reproducibility of PELDI-MS offered a good linear response ($R^2 = 0.963$ - 0.986) with a limit of detection (LOD) of 0.41-0.53 μM in metabolite analysis (Fig 2I and Appendix Fig S3F and G)." (Page 5, Manuscript)

"For analytical techniques, the PELDI-MS showed the advantages of simple sample treatment, fast analytical speed, and low test cost compared with the other typical MS techniques like LC/GC-MS. For simple sample treatment and fast analytical speed, the high salt and protein tolerance of the tailored particles in PELDI-MS allowed direct detection of metabolites in serum (with 60-80 mg/mL of proteins and 135-145 mM of Na^+) free of sample treatment, therefore afford a fast analytical speed of ~30 seconds per sample. The LC/GC-MS require deproteinization and LC/GC in sample treatment to purify and enrich metabolites, with an analytical speed of ~30-60 minutes (Cao *et al*, 2020; Chen *et al*, 2022; Sato *et al*, 2022). For low test cost, the test cost of the PELDI-MS was ~3 dollars, considering all consumables (*e.g.*, ferric oxide particles and standard metabolites for calibration) and equipment depreciation (*e.g.*, laser generator and mass detector) (Chen *et al*, 2023). In comparison, the costs of LC/GC MS were usually ~tens of dollars, with the additional reagents (*e.g.*, reagents for

deproteinization) and instruments (e.g., chromatographic instrument) for sample treatment (Sato *et al.*, 2022). Further, the on-chip microarray design in NPELDI-MS allowed the automatic detection of metabolites with high reproducibility (CVs of 5.6-11.0%), facile for potential large-scale tests in clinics." (Page 12, Manuscript)

"The limit of detection (LOD) was calculated using the following formula: $LOD = 3 \sigma/S$ (Hu *et al.*, 2023; Kim *et al.*, 2023). Where σ is the standard deviation of the method and S is the slope of the calibration curve." (Page 20, Manuscript)

"I. The PELDI-MS offered a good linear response ($R^2 = 0.986$) with a limit of detection (LOD) of 0.50 μM in proline analysis. Data were mean \pm SD, $N = 3$ technical replicates." (Page 30, Manuscript, Figure 2I)

"F, G. The PELDI-MS offered a good linear response ($R^2 = 0.963-0.973$) with a limit of detection (LOD) of 0.41-0.53 μM in (F) alanine and (G) glucose analysis. Data were mean \pm SD, $N = 3$ technical replicates." (Page 4, Appendix, Appendix Figure S3F and G)

Comment 5: Choice of Classification Model: The paper deviates from the common practice of employing Partial Least Squares Discriminant Analysis (PLS-DA), which is widely used in metabolomics, opting instead for various regression-based models for classification. The reasoning behind this unconventional choice should be provided for enhanced clarity.

Response:

We thank the reviewer for the valuable comments. We chose the regression-based models like least absolute shrinkage and selection operator (LASSO) for classification, considering model performance, model robustness, and feature selection.

For model performance, LASSO afforded better performance ($P < 0.05$) with an AUC of 0.957 (95% confidence interval (CI) of 0.906-1.000), compared to PLS-DA (AUC of 0.940 and 95% CI of 0.877-0.996) and other commonly used machine learning algorithms (AUC of 0.757-0.940 for logistic regression, random forest, and decision tree).

For model robustness, our dataset contained small n (n as the sample number) and large p (p as the m/z feature number), which increased the risk of overfitting. LASSO mitigated this risk of overfitting by applying an L_1 -penalty to the coefficients of m/z features, thus enhancing the robustness and generalizability of the constructed model

(Tibshirani *et al*, 2010; Zou & Hastie, 2005). Notably, we also confirmed that there was no overfitting of LASSO model, based on the permutation test ($P < 0.001$, Fig R4A) and the consistent result in an independent validation cohort (AUC of 0.957 and 95% CI of 0.920-0.995), compared with the discovery cohort (AUC of 0.957 and 95% CI of 0.906-1.000).

For feature selection, LASSO is widely used in scenarios with high-dimensional datasets, like metabolomics (Xiao *et al*, 2022; Yao *et al*, 2023). LASSO shrank the coefficients of less informative features towards 0 by applying an L_1 -penalty, thus effectively identifying and retaining only the most relevant features. The optimized L_1 -penalty of 0.6 in LASSO offered 81 m/z features for EC diagnosis with the best AUC of 0.957 (95% CI of 0.906-1.000) in the discovery cohort (Fig R4B). It effectively identified a subset of m/z features with the strongest contributions to the classification task for further biomarker identification, a key aspect when translating the model into potential clinical applications.

Figure R4. (A) The permutation test with 5000 randoms confirmed no overfitting of LASSO model ($P < 0.001$). (B) Optimization of the L_1 -penalty for LASSO model in the discovery cohort. The optimized L_1 -penalty ($L_{opt} = 0.6$) and m/z feature number ($No_{opt} = 81$) were marked with a red dashed line.

We have added the related results and discussion in the manuscript and appendix as follows (Appendix Fig S5B and C and Appendix Table S4).

Revisions:

"We included 5 commonly used machine learning algorithms (least absolute shrinkage and selection operator (LASSO), logistic regression, partial least squares discriminant analysis (PLS-DA), random forest, and decision tree). All algorithms achieved an $AUC \geq 0.75$ in the discovery cohort, demonstrating the potential of SMFs

in EC diagnosis. Specifically, LASSO afforded the best performance ($P < 0.05$) with an AUC of 0.957 (95% confidence interval (CI) of 0.906-1.000, Fig 4B), compared to the other machine learning algorithms (AUC of 0.940 and 95% CI of 0.876-0.998 for logistic regression, AUC of 0.940 and 95% CI of 0.877-0.996 for PLS-DA, AUC of 0.905 and 95% CI of 0.822-0.986 for random forest, AUC of 0.757 and 95% CI of 0.635-0.879 for decision tree, Appendix Table S4)." (Page 7, Manuscript)

"LASSO is widely used in scenarios with high-dimensional datasets, like metabolomics with small n (n as the sample number) and large p (p as the m/z feature number) (Xiao *et al*, 2022; Yao *et al*, 2023a). LASSO mitigated this risk of overfitting by applying an L_1 -penalty to select the most relevant m/z features for classification, thus enhancing the robustness and generalizability of the constructed model (Tibshirani *et al*, 2010; Zou & Hastie, 2005). The optimized L_1 -penalty of 0.6 in LASSO offered 81 m/z features for EC diagnosis with the best AUC of 0.957 (95% CI of 0.906-1.000) in the discovery cohort (Appendix Fig S5B). Notably, we also confirmed that there was no overfitting of LASSO model, based on the permutation test ($P < 0.001$, Appendix Fig S5C) and the consistent result in an independent validation cohort (AUC of 0.957 and 95% CI of 0.920-0.995, Fig 4C), compared with the discovery cohort (AUC of 0.957 and 95% CI of 0.906-1.000)." (Page 8, Manuscript)

"B. Optimization of the L_1 -penalty for LASSO model in the discovery cohort. The optimized L_1 -penalty ($L_{opt} = 0.6$) and m/z feature number ($No_{opt} = 81$) were marked with a red dashed line. C. The permutation test with 5000 randoms confirmed no overfitting of LASSO model ($P < 0.001$)." (Page 6, Appendix, Appendix Figure S5B and C)

Comment 6: Addressing Model Overfitting: The paper overlooks the potential issue of model overfitting, given the higher number of analyzed features in comparison to the sample size. To mitigate this risk, introducing a feature selection strategy, such as the Boruta algorithm or genetic algorithm, could prove useful. Furthermore, it is essential to conduct an overfitting estimation, such as a permutation test, to account for the imbalance between features and observations.

Response:

We are very grateful for the comments. We have introduced a model-based feature selection strategy of LASSO and optimized the selected feature number. Further, we also confirmed that there was no overfitting, based on a permutation test ($P < 0.001$)

and the consistent result in an independent validation cohort (AUC of 0.957 and 95% CI of 0.920-0.995), compared with the discovery cohort (AUC of 0.957 and 95% CI of 0.906-1.000).

We have used the LASSO model for feature selection. LASSO shrank the coefficients of less informative features towards 0 by applying an L_1 -penalty, thus effectively identifying and retaining only the most relevant features. The optimized L_1 -penalty of 0.6 in LASSO offered 81 m/z features for EC diagnosis with the best AUC of 0.957 (95% CI of 0.906-1.000) in the discovery cohort (Fig R5A). This approach not only helps in mitigating overfitting by reducing the model complexity but also enhances interpretability by selecting the most relevant m/z features for further biomarker identification.

Furthermore, we also confirmed that there was no overfitting of LASSO model, based on a permutation test ($P < 0.001$, Fig R5B) and the consistent result in an independent validation cohort (AUC of 0.957 and 95% CI of 0.920-0.995), compared with the discovery cohort (AUC of 0.957 and 95% CI of 0.906-1.000).

Figure R5. A) Optimization of the L_1 -penalty for LASSO model in the discovery cohort. The optimized L_1 -penalty ($L_{opt} = 0.6$) and m/z feature number ($No_{opt} = 81$) were marked with a red dashed line. **B)** The permutation test with 5000 randoms confirmed no overfitting of LASSO model ($P < 0.001$).

We have added the related results and discussion in the manuscript and appendix as follows (Appendix Fig S5B and C).

Revisions:

"LASSO is widely used in scenarios with high-dimensional datasets, like metabolomics with small n (n as the sample number) and large p (p as the m/z feature

number) (Xiao *et al*, 2022; Yao *et al*, 2023a). LASSO mitigated this risk of overfitting by applying an L_1 -penalty to select the most relevant m/z features for classification, thus enhancing the robustness and generalizability of the constructed model (Tibshirani *et al*, 2010; Zou & Hastie, 2005). The optimized L_1 -penalty of 0.6 in LASSO offered 81 m/z features for EC diagnosis with the best AUC of 0.957 (95% CI of 0.906-1.000) in the discovery cohort (Appendix Fig S5B). Notably, we also confirmed that there was no overfitting of LASSO model, based on the permutation test ($P < 0.001$, Appendix Fig S5C) and the consistent result in an independent validation cohort (AUC of 0.957 and 95% CI of 0.920-0.995, Fig 4C), compared with the discovery cohort (AUC of 0.957 and 95% CI of 0.906-1.000)." (Page 8, Manuscript)

"B. Optimization of the L_1 -penalty for LASSO model in the discovery cohort. The optimized L_1 -penalty ($L_{opt} = 0.6$) and m/z feature number ($No_{opt} = 81$) were marked with a red dashed line. C. The permutation test with 5000 randoms confirmed no overfitting of LASSO model ($P < 0.001$)." (Page 6, Appendix, Appendix Figure S5B and C)

Comment 7: Age Discrepancy Reporting: The paper presents an inconsistency in age reporting. While Table S3 indicates a significant age difference ($p < 0.001$) between the studied groups, the Results section suggests that the mean age of cases and controls in the training group does not differ ($p > 0.05$). This inconsistency requires clarification and explanation.

Response:

We are very grateful for the comments. Table S3 showed a significant age difference ($P < 0.001$) between the studied groups. However, we have matched the age distribution ($P > 0.05$) between the EC and Non-EC groups in the training group to mitigate age bias during the training phase and panel development.

We have added the related clarifications and revised the manuscript as follows.

Revisions:

"Notably, we have matched the age distribution ($P > 0.05$) between the EC and Non-EC groups in the discovery cohort to mitigate age bias during the training phase and panel development." (Page 7, Manuscript)

References

- Cao J, Shi X, Gurav DD, Huang L, Su H, Li K, Niu J, Zhang M, Wang Q, Jiang M *et al* (2020) Metabolic fingerprinting on synthetic alloys for medulloblastoma diagnosis and radiotherapy evaluation. *Adv Mater* 32: e2000906
- Chen F, Dai X, Zhou CC, Li KX, Zhang YJ, Lou XY, Zhu YM, Sun YL, Peng BX, Cui W (2022) Integrated analysis of the faecal metagenome and serum metabolome reveals the role of gut microbiome-associated metabolites in the detection of colorectal cancer and adenoma. *Gut* 71: 1315-1325
- Chen W, Yu H, Hao Y, Liu W, Wang R, Huang Y, Wu J, Feng L, Guan Y, Huang L *et al* (2023) Comprehensive metabolic fingerprints characterize neuromyelitis optica spectrum disorder by nanoparticle-enhanced laser desorption/ionization mass spectrometry. *ACS Nano* 17: 19779-19792
- Hu FX, Hu G, Wang DP, Duan X, Feng L, Chen B, Liu Y, Ding J, Guo C, Yang HB (2023) Integrated biochip-electronic system with single-atom nanozyme for in vivo analysis of nitric oxide. *ACS Nano* 17: 8575-8585
- Huang B, Song BL, Xu C (2020) Cholesterol metabolism in cancer: mechanisms and therapeutic opportunities. *Nat Metab* 2: 132-141
- Ihata Y, Miyagi E, Numazaki R, Muramatsu T, Imaizumi A, Yamamoto H, Yamakado M, Okamoto N, Hirahara F (2014) Amino acid profile index for early detection of endometrial cancer: verification as a novel diagnostic marker. *Int J Clin Oncol* 19: 364-372
- Kim JY, Koh EH, Yang JY, Mun C, Lee S, Lee H, Kim J, Park SG, Kang M, Kim DH *et al* (2023) 3D plasmonic gold nanopocket structure for SERS machine learning-based microplastic detection. *Adv Funct Mater*
- Liu W, Luo Y, Dai J, Yang L, Huang L, Wang R, Chen W, Huang Y, Sun S, Cao J *et al* (2022) Monitoring retinoblastoma by machine learning of aqueous humor metabolic fingerprinting. *Small Methods* 6: e2101220
- Miyagi E, Maruyama Y, Mogami T, Numazaki R, Ikeda A, Yamamoto H, Hirahara F (2016) Comparison of plasma amino acid profile-based index and CA125 in the diagnosis of epithelial ovarian cancers and borderline malignant tumors. *Int J Clin Oncol* 22: 118-125
- Nava Lauson CB, Tiberti S, Corsetto PA, Conte F, Tyagi P, Machwirth M, Ebert S, Loffreda A, Scheller L, Sheta D *et al* (2023) Linoleic acid potentiates CD8(+) T cell metabolic fitness and antitumor immunity. *Cell Metab* 35: 633-650
- Sato S, Dyar KA, Treebak JT, Jepsen SL, Ehrlich AM, Ashcroft SP, Trost K, Kunzke T, Prade VM, Small L *et al* (2022) Atlas of exercise metabolism reveals time-dependent signatures of metabolic homeostasis. *Cell Metab* 34: 329-345

Tibshirani R, Hastie T, Friedman J (2010) Regularized paths for generalized linear models via coordinate descent. *Journal of Statistical Software* 33: 1

Virtanen P, Gommers R, Oliphant TE, Haberland M, Reddy T, Cournapeau D, Burovski E, Peterson P, Weckesser W, Bright J *et al* (2020) SciPy 1.0: fundamental algorithms for scientific computing in Python. *Nat Methods* 17: 261-272

Xiao Y, Ma D, Yang YS, Yang F, Ding JH, Gong Y, Jiang L, Ge LP, Wu SY, Yu Q *et al* (2022) Comprehensive metabolomics expands precision medicine for triple-negative breast cancer. *Cell Res* 32: 477-490

Yao Y, Wang X, Guan J, Xie C, Zhang H, Yang J, Luo Y, Chen L, Zhao M, Huo B *et al* (2023) Metabolomic differentiation of benign vs malignant pulmonary nodules with high specificity via high-resolution mass spectrometry analysis of patient sera. *Nat Commun* 14: 2339

Yu W, Wu B, Lin N, Stone K, Williams K, Zhao H (2006) Detecting and aligning peaks in mass spectrometry data with applications to MALDI. *Comput Biol Chem* 30: 27-38

Zou H, Hastie T (2005) Regularization and variable selection via the elastic net. *J Royal Stat Soc B* 67: 301-320

19th Jan 2024

Dear Prof. Kun,

Thank you for submitting your revised manuscript, and please accept my apologies for the delay in getting back to you in this busy time of the year. Unfortunately referee #2 was not able to assess the revised manuscript, however referee #1 has evaluated your responses to both referees' concerns. As you will see below, this referee is satisfied with the revisions, and I will therefore be able to accept your manuscript once the following editorial points will be addressed:

1/ Manuscript text:

- Please remove the yellow highlights, and only keep in track changes mode any new modification.
- Please provide up to 5 keywords
- Materials and Methods: please indicate whether the cells were authenticated.
- Data availability: As per our guidelines, large-scale datasets, sequences, atomic coordinates and computational models should be deposited in one of the relevant public databases prior to submission. Accession codes should be included in a "Data Availability" section at the end of Materials & Methods (suggested wording: "The [protein interaction | microarray | mass spectrometry] data from this publication have been deposited to the [name of the database] database [URL] and assigned the identifier [accession | permalink | hashtag].")
In particular, mass spectrometry datasets should be deposited in a machine-readable format (e.g. mzML if possible) in one of the major public database, for example Pride or PeptideAtlas and authors should follow the MIAPE recommendations. Microarray and sequencing-based functional genomics data should be deposited in the ArrayExpress or GEO databases in compliance to the MIAME standards and the MINSEQE draft proposal.

2/ Figures and Appendix:

- Please provide exact p values, not a range, in the figures or their legends.
- The figures should be removed from the manuscript text, and uploaded as EPS, TIFF or PDF format.
- Figure legends: Although 'n' is provided, please describe the nature of entity for 'n' in the legends of figures 5 c,e.
- Appendix: please remove the yellow highlights and correct the figure callouts to Appendix Figure S1, etc.
- Please note that you have the possibility to replace Supplementary Information with Expanded View (EV) Figures and Tables that are collapsible/expandable online. A maximum of 5 EV Figures can be typeset. EV Figures should be cited as 'Figure EV1, Figure EV2' etc... in the text and their respective legends should be included in the main text after the legends of regular figures. See detailed instructions here:

3/ Checklist:

- please complete the section on statistics/inclusion and exclusion criteria.
- please double check the section Ethics/specimen and field samples, as I don't think it applies in your case.
- please complete the section Data availability/ Primary datasets deposition.

4/ The Paper Explained: I added minor edits to your text, please let me know if you agree or amend as you see fit:
Problem

Endometrial cancer (EC) is the most prevalent gynaecological cancer worldwide. Existing diagnostic tools, including transvaginal ultrasound, biopsy, and curettage, are limited in terms of specificity (~51.1% for transvaginal ultrasound) and invasiveness (endometrial sampling for biopsy and curettage). Moreover, the commonly used blood biomarker for EC diagnosis, cancer antigen 125 (CA-125), exhibits low sensitivity (< 60%).

Results

We recorded the high-performance serum metabolic fingerprints (SMFs) of 191 EC and 204 Non-EC subjects via particle-enhanced laser desorption/ionization mass spectrometry (PELDI-MS). Further, we identified a metabolic biomarker panel of glutamine, glucose, and cholesterol linoleate with an AUC of 0.901-0.902 and an accuracy of 82.8-83.1% for EC diagnosis through machine learning of SMFs. Finally, we validated the function of the 3 metabolite biomarkers on EC cell behaviour, including proliferation, colony formation, migration, and apoptosis in vitro.

Impact

Our work will facilitate the development of novel diagnostic biomarkers for EC. Functional validation of these metabolite biomarkers provides biological insights for their use as diagnostic biomarkers.

5/ I slightly edited your synopsis text to match our style and format, please let me know if you agree or amend as you see fit:

Endometrial cancer (EC) diagnostic suffers from a lack of non-invasive, specific and sensitive tools. Machine learning of high-performance serum metabolic fingerprints (SMFs) was used to identify a metabolic biomarker panel for differentiation diagnosis of EC vs. Non-EC.

- An SMFs database of 191 EC and 204 Non-EC subjects was built via particle-enhanced laser desorption/ionization mass spectrometry (PELDI-MS).

- A metabolic biomarker panel for differentiation diagnosis of EC was identified, with an AUC of 0.901-0.902 and an accuracy of 82.8-83.1%.

- The metabolite biomarker functions on EC cell behaviour were evaluated in vitro (including proliferation, colony formation, migration, and apoptosis).

6/ As part of the EMBO Publications transparent editorial process initiative (see our Editorial at <http://embomolmed.embopress.org/content/2/9/329>), EMBO Molecular Medicine will publish online a Review Process File (RPF) to accompany accepted manuscripts.

This file will be published in conjunction with your paper and will include the anonymous referee reports, your point-by-point response and all pertinent correspondence relating to the manuscript. Let us know whether you agree with the publication of the RPF and as here, if you want to remove or not any figures from it prior to publication.

I look forward to receiving your revised manuscript.

Yours sincerely,

Lise Roth

***** Reviewer's comments *****

Referee #1 (Remarks for Author):

Authors have addressed the concerns. The manuscript can be accepted as it.

The authors addressed the editorial issues.

26th Jan 2024

Dear Prof. Kun,

Thank you for submitting your revised files. I am pleased to inform you that your manuscript is accepted for publication and is now being sent to our publisher to be included in the next available issue of EMBO Molecular Medicine!

If you have any questions, please do not hesitate to contact the Editorial Office.

Congratulations on your interesting work!

Yours sincerely,

Lise Roth
